# A Celsr3 Mutation Linked to Tourette Disorder Disrupts Cortical Dendritic Patterning and Striatal Cholinergic Interneuron Excitability

**DOI:** 10.3390/ijms262110307

**Published:** 2025-10-23

**Authors:** Cara Nasello, G. Duygu Yilmaz, Lauren A. Poppi, Tess F. Kowalski, K. T. Ho-Nguyen, Junbing Wu, Matthew Matrongolo, Joshua K. Thackray, Anna Shi, Nicolas L. Carayannopoulos, Nithisha Cheedalla, Julianne McGinnis, Jasmine Chen, Adyan Khondker, Fadel Tissir, Gary A. Heiman, Jay A. Tischfield, Max A. Tischfield

**Affiliations:** 1Department of Cell Biology and Neuroscience, Rutgers, The State University of New Jersey, Piscataway, NJ 08854, USA; cnasello@hginj.rutgers.edu (C.N.); gdy6@dls.rutgers.edu (G.D.Y.); tfk28@dls.rutgers.edu (T.F.K.); kt.honguyen.a@gmail.com (K.T.H.-N.); junbing.wu@icloud.com (J.W.); mjm4@stanford.edu (M.M.); thackray@hginj.rutgers.edu (J.K.T.); as2800@scarletmail.rutgers.edu (A.S.); nlc90@rwjms.rutgers.edu (N.L.C.); nithisha.cheedalla@gmail.com (N.C.); jem412@rwjms.rutgers.edu (J.M.); jc3002@scarletmail.rutgers.edu (J.C.); aak283@scarletmail.rutgers.edu (A.K.); 2Keck Center for Collaborative Neuroscience, Rutgers, The State University of New Jersey, 604 Allison Road, D251, Piscataway, NJ 08854, USA; 3Department of Genetics and the Human Genetics Institute of New Jersey, Rutgers, The State University of New Jersey, Piscataway, NJ 08854, USA; laurenpoppi90@gmail.com (L.A.P.); heiman@rutgers.edu (G.A.H.); jay@hginj.rutgers.edu (J.A.T.); 4Child Health Institute of New Jersey, Robert Wood Johnson Medical School, New Brunswick, NJ 08901, USA; 5College of Health and Life Sciences, Hamad Bin Khalifa University, Doha 34110, Qatar; fadel.tissir@uclouvain.be; 6Laboratory of Developmental Neurobiology, Institute of Neuroscience, Université Catholique de Louvain, 1200 Brussels, Belgium

**Keywords:** Tourette, Celsr3, interneuron, sensorimotor, cortex, striatum, grey matter

## Abstract

Tourette Disorder (TD) is a prevalent neurodevelopmental condition characterized by chronic motor and vocal tics. A mechanistic understanding of both the genetic etiology and brain pathophysiology remains poor. To gain insight into the molecular underpinnings of TD, we have generated a novel mouse model expressing an orthologous human mutation in *CELSR3*, a high-confidence TD risk gene. This putative damaging de novo variant, R774H, causes an amino acid substitution within the fifth cadherin repeat. Unlike previous *Celsr3* TD models and *Celsr3* constitutive null mice, mice homozygous for the R774H amino acid substitution are viable. They have grossly normal forebrain development and no changes to the density of cortical and striatal interneuron subpopulations. However, 3D geometric analysis of cortical pyramidal neurons revealed changes to dendritic patterning and the types and distributions of spines. Furthermore, patch clamp recordings in cholinergic interneurons located within the sensorimotor striatum uncovered mild intrinsic hyperexcitability and changes to spine density. Despite these changes, *Celsr3^R774H^* homozygous mice do not show repetitive motor behaviors at baseline nor motor learning impairments. However, *Celsr3^R774H^* homozygous males have sensorimotor gating deficits, a behavioral phenotype observed in both humans with TD and previously reported mouse models. Our findings suggest human mutations in CELSR3 may affect dendritic patterning, spine formation and/or turnover, and the firing properties of neurons within cortico-striatal circuits.

## 1. Introduction

Tourette Disorder (TD) is a childhood-onset neurodevelopmental disorder characterized by chronic vocal and motor tics, which are often elicited in response to somatosensory phenomena known as premonitory sensations [1]. TD manifests with neuropsychiatric comorbidities including attention-deficit hyperactivity disorder, obsessive–compulsive disorder, autism spectrum disorder, as well as mood, anxiety, and sleep disorders, which altogether underscore its multifaceted pathophysiology [2,3,4,5]. TD is a heritable disorder, with a concordance rate approaching 80% in monozygotic twins [6]. Human imaging studies have revealed structural and functional changes spanning cortico-striatal-thalamo-cortical (CSTC) and basal ganglia networks that govern the selection, planning, and control of volitional behavioral output [7,8,9,10,11].

Structural and functional changes in sensorimotor control loops likely contribute to the pathology of TD [12,13,14,15]. Wide-spread alterations in cortical thickness (grey matter) and intercortical innervation patterns (white matter) have been reported in TD subjects, although findings vary. Both increased and decreased grey matter volumes have been observed in the somatosensory and motor cortices, the putamen, thalamus, hypothalamus, and midbrain nuclei [16,17,18,19,20]. Functional magnetic resonance imaging consistently shows increased activity in these areas and a stronger connectivity between the sensorimotor cortex and the putamen, suggesting a heightened descending drive from sensorimotor areas [10,21,22]. In contrast, studies of higher-order associative cortices show mostly decreased grey matter volume and diminished functional activity in orbitofrontal, prefrontal, parietal, occipital and premotor cortices [16,21,23,24]. Increased insular, hippocampal and amygdalar volumes in patients with strong premonitory sensations may be structural consequences of tic regulation [20,25,26]. Given TD’s developmental trajectory, determining whether these structural changes are causal or compensatory is challenging.

These structural differences can be due to microscopic changes to neural circuits. For example, a change in cortical thickness might reflect underlying changes to the neuropil (dendritic arborization, synaptogenesis, glial processes), to neuronal composition and distribution, or to myelination patterns within surrounding white matter. Immunohistochemical approaches on post-mortem brain tissue provide a more detailed picture, but these findings are limited to the striatum. Post-mortem brain tissue of subjects with severe, refractory TD has revealed signs of parvalbumin and cholinergic interneuron loss in striatal regions and may account for tics and related behaviors in at least some affected individuals [27,28,29,30]. Focal disinhibition of the dorsal sensorimotor striatum via application of GABA_A_ receptor antagonists in rodents and non-human primates can trigger severe motor stereotypies [31,32,33], whereas disinhibition in the ventral striatum leads to vocalizations [34,35]. Furthermore, targeted ablation of cholinergic interneurons in the rodent dorsal striatum causes motor stereotypies as well as perseverative behaviors following acute stress or amphetamine challenge [36,37]. While these animal models have provided valuable insights for understanding the pathogenesis of TD, they have notable limitations as the experimental manipulations were performed in adult animals and likely fail to model the types of neurodevelopmental changes found in humans.

Historically, the limited availability of preclinical genetic models for TD has hindered the development of targeted treatment [14]. Despite the prevalence of TD (~0.5–1% of the population) [38], gene coding variants have been identified in relatively few families, and genome-wide association studies have yielded few clues, often failing to replicate in subsequent studies [39]. However, recurrent de novo coding variants in several genes, including *CELSR3*, *NIPBL*, and *WWC1*, have been identified using large scale trio based whole-exome sequencing [14,40]. Of the identified high-confidence candidates, multiple likely damaging or loss-of-function mutations in several domains of *CELSR3* have been identified in simplex trios and multiplex families [41].

*CELSR3* encodes a protocadherin cell adhesion G protein-coupled receptor that is critical for axon guidance, dendritic patterning, and the formation of synapses. *CELSR3* is required for the development and guidance of major forebrain axon tracts, such as the anterior commissure and internal capsule, which contains corticostriatal, thalamocortical, and corticothalamic axons [42,43,44]. During development, *CELSR3* is required for the establishment of basal ganglia pathways, including axonal projections from areas such as the striatum, subthalamic nucleus, and the substantia nigra pars compacta to the globus pallidus [45]. Mice homozygous for a GFP knock-in allele (ablating *Celsr3* expression) suggest Celsr3 may be required for the tangential migration of interneurons, although these findings have not been replicated in other *Celsr3* mutant models [46,47]. In adult animals, *Celsr3* expression is maintained in subpopulations of cortical and striatal interneurons, as well as cerebellar Purkinje neurons [48]. Thus, *Celsr3* expression patterns in the brain and its necessity for the development of CSTC and basal ganglia circuitry make it an attractive candidate to model TD in mice.

We previously reported that mice expressing orthologous human mutations within the second laminin G-like domains of *Celsr3* show sensorimotor gating deficits, repetitive motor behaviors, and changes to reward learning and electrically evoked striatal dopamine release [49]. In the present study, we have developed a novel model for TD that expresses an orthologous human amino acid substitution, R774H (in humans, R783H), within the fifth protocadherin repeat of the extracellular domain of *Celsr3*. We investigated the impact of the *Celsr3*^R774H^ amino acid substitution on mouse behavior and brain development. We hypothesized that *Celsr3*^R774H^ mutant mice would show similar behavioral changes to those previously described in other *Celsr3* models of TD [49] as well as changes to axon guidance and/or dendritic patterning. Unlike our previously reported models, mice heterozygous for the R774H amino acid substitution do not show behavioral changes in the paradigms tested. However, males homozygous for the R774H amino acid substitution have sensorimotor gating deficits that are absent in affected females, while homozygous females, but not males, have increased marble burying, demonstrating sex-specific phenotypes in this model. In agreement with our previous findings, we do not see evidence of either cortical or striatal interneuron loss in homozygous mutants, and the development of major white matter tracts in the forebrain also appears grossly normal. However, we find subtle changes to dendritic patterning and synapse formation in deep layer cortical pyramidal neurons, as well as changes to striatal cholinergic interneuron excitability. Our findings demonstrate that human mutations in *CELSR3* are sufficient to cause discernible changes to neurite development and synapse formation, supporting a framework in which impairments in the ability of neurons to functionally integrate into CSTC loops might underlie TD.

## 2. Results

*Celsr3^R774H/+^* and *Celsr3^R774H/R774H^* animals on a pure C57BL/6 background were born at normal Mendelian ratios, had normal weights, and were indistinguishable from littermate controls (no hair loss or skin lesions). By contrast, we previously reported that homozygous *Celsr3^C1906Y^* and *Celsr3^S1894Qfs*2^* mice are perinatal lethal, like *Celsr3* null mice, suggesting the *Celsr3^R774H^* mutant protein retains partial function (Figure 1a). Furthermore, protein levels in whole-brain lysates were normal in homozygous *Celsr3^R774H^* neonates, by contrast with previous models (Figure 1b). Thus, the R774H substitution in the 5th protocadherin repeat appears to exert milder effects versus those found in the second laminin G-like domain.

### 2.1. Celsr3^R774H/R774H^ Mice Show Sex-Specific Sensorimotor Gating Deficits

TD subjects show deficits in sensorimotor gating as measured by prepulse inhibition (PPI) of the acoustic or tactile startle reflex [50,51]. Similarly, several genetic mouse models of TD have sensorimotor gating deficits [49,52,53]. These findings suggest that perturbations to sensorimotor gating may be useful as a behavioral screening test to validate animal models of TD. Therefore, we tested sensorimotor gating in *Celsr3*^R774H^ mice using an acoustic PPI paradigm. All *Celsr3^R774H^* mice showed comparable levels of baseline acoustic startle reflex as wild-type control littermates (Figure 1c). Both male and female *Celsr3^R774H/+^* mice showed no differences in PPI of acoustic startle (Figure 1d,e). However, PPI was mildly, yet significantly, reduced in male *Celsr3^R774H/R774H^* mice compared to wild-type littermate controls (Sidak’s multiple comparisons testing, pp71: *p* = 0.6, pp77: *p* = 0.03 and pp81, *p* = 0.06, Figure 1d). Female *Celsr3*^R774H/R774H^ mice did not show significant PPI deficits at any of the dB prepulses tested (Figure 1e). Due to our findings showing changes only in homozygous mice, we restricted our focus to *Celsr3^R774H/R774H^* animals for subsequent experiments.

### 2.2. Celsr3^R774H/R774H^ Mice Do Not Show Hyperactivity or Repetitive Motor Behaviors in the Open Field

*Celsr3^C1906Y/+^* and *Celsr3^S1894Rfs*/+^* mice have increased locomotion and repetitive rearing in an open field arena; therefore, we tested *Celsr3^R774H/R774H^* mice and their wild-type littermates for similar behavioral changes. *Celsr3^R774H/R774H^* male and female mice did not show an increase in overall activity compared to wild-type littermate controls (Figure 2a,b). However, males trended upward suggesting a milder—but shared—increased activity phenotype with *Celsr3^C1906Y/+^* and *Celsr3^S1894Rfs*/+^* mutants (Figure 2b, *p* = 0.16, 2-way RM-ANOVA). Next, we looked at the number of rearing events and time spent rearing. The number of events and amount of time spent rearing were equivalent between *Celsr3^R774H/R774H^* male and female mice versus wild-type littermates (Figure 2c). We next looked at changes in the time spent in the center of the arena, a measure of exploratory behavior and anxiety (Figure 2d). *Celsr3^R774H/R774H^* male and female mice spent equivalent amounts of time in the center compared to wild-type littermate controls. However, both male and female *Celsr3^R774H/R774H^* mice spent more time in the center during the third 10 min testing period compared to the first 10 min period, suggesting changes to exploratory behavior and/or anxiety levels as time progressed. (Figure 2d Sidak’s multiple comparisons, male: *p* = 0.044 female, *p* = 0.049, 2-way RM-ANOVA).

### 2.3. Celsr3^R774H/R774H^ Mice Exhibit Normal Motor Coordination as Measured by Accelerated Rotarod

Mutations in *Celsr3* are reported to alter rotarod performance in mice. Mice with conditional knockout of *Celsr3* within Purkinje cells in the cerebellum have worse performance in the rotarod, whereas our previously reported *Celsr3* TD models perform better [48,49]. Therefore, we tested balance and motor coordination using an accelerated rotarod protocol. *Celsr3^R774H/R774H^* male and female mice showed comparable latency to fall as wild-type littermate controls (males: *p* = 0.76, females: *p* = 0.62, 2-way RM-ANOVA), suggesting motor coordination and learning is normal and intact in these animals (Figure 2e).

### 2.4. Female Celsr3^R774H/R774H^ Mice Show Perseverative Digging Behavior

It has been previously reported that mutations in *Celsr3* mice can cause sex-specific changes to perseverative digging behavior [49,54]. We examined perseverative behaviors using the marble burying assay, a test of repetitive digging behavior (Figure 2f). Male *Celsr3^R774H/R774H^* mice buried similar numbers of marbles compared to wild-type littermates; however, female *Celsr3^R774H/R774H^* mice buried a significantly higher number of marbles on average compared to littermate controls (*p* = 0.013; two-tailed *t*-test), suggesting sex-specific changes to object-oriented perseverative behavior.

### 2.5. Axon Tract Development Is Grossly Normal in Celsr3^R774H/R774H^ Mice

The gross anatomy and overall size of *Celsr3^R774H/R774H^* mouse brains appeared normal. Given the central role Celsr3 has on axonal tract development in the forebrain, we evaluated the major CSTC pathways (Figure 3a). Antibody labelling against neuronal cell adhesion protein L1 in embryonic day (E) 18.5 brain sections showed that the development and trajectories of major forebrain axon tracts in the internal capsule, anterior commissure, and corpus callosum were grossly normal in *Celsr3^R774H/R774H^* mice (Figure 3b). By comparison, these tracts are absent in *Celsr3* null mutants [47]. Striatonigral axons in the direct pathway terminating in the globus pallidus internus and substantia nigra were visualized by crossing *Drd1a-Cre* and *R26*:*Ai14* reporter lines (Appendix A). Using widefield fluorescence microscopy, striatonigral fiber tracts showed normal development and terminated appropriately in the globus pallidus internus in adult *Drd1a-Cre*;*Celsr3^R774H/R774H^*;*R26*:*Ai14* mice (Figure 3c). Next, we crossed *A2a*-Cre and *R26*:*Ai14* reporter lines (Appendix A) to visualize striatopallidal axons that terminate in the globus pallidus externus. We also did not detect any major qualitative differences in the pattern of tdTomato-positive fibers terminating in the globus pallidus externus of *A2A-Cre*;*Celsr3^R774H/R774H^*;*R26*:*Ai14* animals compared to littermate controls. By contrast, both striatopallidal and striatonigral axon tracts fail to form in mice with constitutive loss of Celsr3 [45]. Axon pathfinding appeared normal as we did not observe instances of wandering axons, bundles, or mis-innervation by direct and indirect pathway terminals (Figure 3c,d). Thus, the *Celsr3^R774H^* amino acid substitution within the fifth cadherin repeat does not affect the ability of the protein to regulate axon guidance in the forebrain in a manner that is detectable with the qualitative anatomical techniques used.

### 2.6. Celsr3^R774H/R774H^ Mice Have Organized Cortical Layering and Do Not Show Interneuron Loss

Cortical layering, as assessed by TBR1, CTIP2, and SATB2 immunostaining, was normal in *Celsr3^R774H/R774H^* animals compared to littermate controls (Figure 4a). The relative radial thickness of each cortical layer was also normal in *Celsr3^R774H/R774H^* animals (Figure 4b), and nearest neighbor (NN) analysis showed normal distribution of labelled cortical neurons (Figure 4b, *p* = 0.2275, 2-way ANOVA). *Celsr3* is expressed by E13.5 in the ganglionic eminences, which give rise to cortical and striatal interneurons, and regulates the tangential migration of cortical interneurons [46]. Immunolabeling against parvalbumin showed that the density of cortical parvalbumin interneurons was normal in *Celsr3^R774H/R774H^* mice, despite variable staining differences that occurred in thick sections from both wild-type and mutant mice (Figure 4d,e). Using *Somatostatin-Cre* and the *R26:Ai14* reporter line to lineage label somatostatin interneurons, there were also no differences in density within the cortex of *Celsr3^R774H/R774H^* mice (Figure 4f,g). Thus, cell proliferation and the radial and tangential migration of cortical pyramidal neurons and interneurons, respectively, appeared normal in *Celsr3^R774H/R774H^* animals.

### 2.7. Cortical Pyramidal Neuron Dendritic Patterning Is Affected in Celsr3^R774H/R774H^ Mice

Celsr3 is required for neurite development and dendritic patterning in the cortex and hippocampus [55,56]. Given broad *Celsr3* expression across cortical PV interneurons, we attempted to examine whether the dendritic arborizations of cortical parvalbumin interneurons were properly patterned in *Celsr3^R774H/R774H^* mice by using a Cre-dependent viral sparse cell labelling approach to mark parvalbumin (PV) interneurons with GFP (Figure 5a,b) [57]. We crossed a *PV-2A-Cre* allele onto the *Celsr3^R774H/R774H^* background and injected the virus into the somatosensory cortex. Most labelled neurons localized to deep layer 5 of the cortex but surprisingly, most were not positive for parvalbumin immunostaining. Instead, these neurons had typical cortical pyramidal neuron morphology with basal and long apical dendrites. Crossing these animals to the *R26*:*Ai14* reporter line showed diffuse td-Tomato expression throughout the cortex, suggesting the *PV-2A-Cre* allele went germline, consistent with previous reports [57].

Nonetheless, 3D neuronal reconstructions revealed that the basal dendrites of *Celsr3^R774H/R774H^* deep layer 5 pyramidal neurons were less arborized than littermate controls (Figure 5b). Basal dendrites were analyzed separately by excluding apical branches from the dataset (Figure 5c). Sholl analysis revealed a genotype effect for the complexity of *Celsr3^R774H/R774H^* pyramidal neuron basal dendrites (Figure 5d *Celsr3^+/+^ n* = 6; *Celsr3^R774H/R774H^ n* = 8; 2-way ANOVA genotype effect *p* < 0.001). The area under the Sholl curve was 1639 ± 40.05 and 1204 ± 27.62 for *Celsr3^+/+^* and *Celsr3^R774H/R774H^*, respectively. There was also a significant genotype effect when comparing branch depth, which reflects the number of times a dendrite has branched since leaving the soma (Figure 5e; *p* = 0.0271, 2-way ANOVA). There was no significant difference in the number of branch points (*Celsr3^+/+^* = 16.33 ± 2.81, *Celsr3^R774H/R774H^* = 15.22 ± 1.52, *p* = 0.7110, unpaired *t*-test) or dendritic straightness (*Celsr3^+/+^* = 0.9411 ± 0.003, *Celsr3^R774H/R774H^* = 0.9298 ± 0.006, *p* = 0.1805, unpaired *t*-test). We also did not see evidence of increased number of self-crossings.

There was no difference in the density of spines along the secondary basal dendrites between *Celsr3^+/+^* (8.60/10 µm) and *Celsr3^R774H/R774H^* (8.94/10 µm) mice (Figure 5f). However, when spines were classified according to morphology (e.g., stubby, mushroom, long-thin, filopodia, see Section A.2 Table A2 for criteria), and the relative densities were compared using the *ClassifySpines* IMARIS plug-in, the proportion of stubby and long-thin spines detected along a single length of dendrite appeared shifted in *Celsr3^R774H/R774H^* mice (Figure 5g). There was also a significant reduction in stubby spines in *Celsr3^R774H/R774H^* animals (*p* = 0.033), and a strong trend toward an increase in long thin spines (*p* = 0.055, multiple *t*-tests corrected for multiple comparisons). Thus, the *Celsr3^R774H^* amino acid substitution in the homozygous state is sufficient to alter dendritic patterning as well as the types and distributions of spines in deep layer cortical pyramidal neurons.

### 2.8. Celsr3^R774H/R774H^ Cholinergic Interneurons Have Altered Membrane Properties and Spine Density

Loss of striatal interneurons, including cholinergic interneurons (CINs), have been reported in adults with severe, refractory TD [28,30,43]. Although we previously reported that the numbers and positioning of striatal parvalbumin interneurons and CINs were normal in *Celsr3^C1906Y/+^* and *Celsr3^S1894Rfs*2/+^* mice [49], we nonetheless asked if mutations in the cadherin domain affected the numbers or positioning of CINs. We crossed *Celsr3^R774H/R774H^* mutants with ChAT-eGFP reporter mice and compared the numbers and distributions of CINs from selected axial positions in the striatum (Figure 6a,b). Consistent with our previous models, we did not see any changes in the numbers (Figure 6c) or distributions (Figure 6d) of CINs.

Although we did not detect signs of striatal interneuron loss in *Celsr3^R774H/R774H^* mice, it is possible that the mutation may alter the active and/or passive membrane properties of these cells. *Celsr3* is widely expressed throughout CINs (Appendix A). By comparison, in a *Celsr3^eGFP/+^* reporter mouse, we observed less colocalization between GFP expressing cells and PV+ interneurons, although their difference failed to reach significance (Appendix A, Wilcoxon signed-rank test, one-tailed, median difference between ChAT-PV: 0.33, *p* = 0.125). Given the significance of CINs in TD literature, we next examined their electrophysiological properties in *Celsr3^R774H/R774H^* mice. CINs located within the dorsolateral striatum of both *Celsr3^+/+^* and *Celsr3^R774H/R774H^* mice had characteristically large somata, a tonic firing profile, with varying levels frequency adaptation between cells, and upon membrane breakthrough, had a relatively depolarized resting membrane potential (RMP, Figure 7a,d). Passive membrane properties were not significantly different between *Celsr3^+/+^* and *Celsr3^R774H/R774H^* mice (Figure 7e–h, see Section A.1 Table A1 for more information). Membrane impedance (R_m_) was 184.8 ± 8.82 MΩ and 205.1 ± 11.89 MΩ for *Celsr3^+/+^* (*n* = 31) and *Celsr3^R774H/R774H^* (*n* = 39) CINs, respectively (*p* = 0.4238, Mann–Whitney test). Membrane capacitance (C_m_) was 33.52 ± 1.10 pF and 34.05 ± 0.97 pF for *Celsr3^+/+^* and *Celrs3^R774H/R774H^* CINs, respectively (*p* = 0.7214, *t*-test). Membrane time constant (tau) was 2.88 ± 0.16 ms and 2.91 ± 0.16 ms for *Celsr3^+/+^* and *Celsr3^R774H/R774H^* CINs, respectively (*p* = 0.8870, *t*-test). Resting membrane potential (RMP) was on average more depolarized in *Celsr3^R774H/R774H^* CINs (*p* = 0.037, *t*-test, Figure 7f). Rheobase (minimum current injection step required to elicit an action potential) was not significantly affected (*p* = 0.3505, Mann–Whitney test, Figure 7i). The action potential (AP) threshold was significantly more depolarized in *Celsr3^R774H/R774H^* CINs (*p* = 0.0456, *t*-test, Figure 7j). The f/I plots for *Celsr3^+/+^* (*n* = 29) and *Celsr3^R774H/R774H^* (*n* = 25) required different nonlinear fits (*p* < 0.001, Figure 7k). This indicated a tendency for *Celsr3^R774H/R774H^* CINs to fire at a higher frequency in response to somatic current injection compared with *Celsr3^+/+^* CINs. AP frequency was significantly higher in *Celsr3^R774H/R774H^* compared to *Celsr3^+/+^* CINs with 200 pA current injection (*p* = 0.038, *t*-test, Figure 7l). Thus, *Celsr3^R774H^* is sufficient to alter the membrane properties of cholinergic interneurons.

In a subset of neurons, we assessed changes to dendrite morphology using biotin filling during recording and post hoc anatomical recovery. *Celsr3^R774H/R774H^* CINs showed changes to neurite complexity compared to *Celsr3*^+/+^ CINs (Sholl analysis, *n* = 6, *p* < 0.001, 2way ANOVA, (Appendix A). The number of branch points trended towards an increase in *Celsr3^R774H/R774H^* CINs (*p* = 0.06, *t*-test, Appendix A), whereas neurite straightness trended towards a decrease in *Celsr3^R774H/R774H^* CINs (*p* = 0.05, *t*-test, Appendix A). In addition, while fractal dimension (D_B_) was similar between *Celsr3*^+/+^ and *Celsr3^R774H/R774H^* CINs (*p* = 0.2636, Mann–Whitney test, Appendix A), lacunarity was significantly increased in *Celsr3^R774H/R774H^* CINs compared to controls (*p* = 0.0379, *t*-test, Appendix A), potentially indicating that dendritic branches are distributed unevenly in mutant cells. Finally, although the distribution of dendritic spines on CINs is much sparser compared to neighboring medium spiny neurons, the average spine density along second order dendrites was decreased in *Celsr3^R774H/R774H^* mice compared to controls (*p* = 0.0184, *t*-test, Figure 7m,n). Given the widespread GFP expression observed in Celsr3^GFP/+^ mice in adult CINs, these observations may reflect cell-autonomous changes resulting from mutant Celsr3, and could account, at least in part, to changes in membrane properties and firing activities. 

## 3. Discussion

In this study, we sought to expand on the available animal models carrying mutations in *Celsr3*, a high-confidence TD gene, by presenting a phenotypic analysis of a novel mouse model engineered to express a putative damaging variant that causes an amino acid substitution within the fifth extracellular cadherin repeat (R774H). This mutation is orthologous to a human variant, R783H, that was identified previously in a whole exome sequencing study [5]. While heterozygous mutants showed no overt behavioral abnormalities, homozygous *Celsr3^R774H/R774H^* males exhibited deficits in sensorimotor gating as measured by prepulse inhibition, whereas homozygous females displayed increased perseverative digging behavior. At the gross anatomical level, we observed no obvious disruptions to major forebrain fiber tracts or changes to interneuron distribution in the cortex and striatum. However, we identified a reduction in the complexity of basal dendritic arbors in deep layer pyramidal neurons located within the primary somatosensory cortex, accompanied by altered dendritic spine distributions. We also observed changes to the membrane properties and firing activities of striatal CINs in ex vivo brain slices, with altered spine density along second order dendrites. Taken together, these results suggest mutations that reside within the extracellular cadherin repeats of *Celsr3* have the potential to disrupt neurite patterning (without gross changes to white matter tracts), synapse formation/turnover, and the firing properties of neurons. These findings highlight the importance of characterizing distinct TD-associated mutations in Celsr3 found across different functional domains to uncover how specific molecular insults may disrupt brain development and contribute to the pathogenesis of TD either through shared or distinct ways. As such, these findings provide a roadmap for extended analyses in *Celsr3^C1906Y/+^* and *Celsr3^S1894Rfs*2/+^* models, both of which show more pronounced behavioral phenotypes, suggesting the anatomical phenotypes observed in the current study may be present, and potentially more severe, in those models as well.

*Celsr3^R774H/R774H^* males have PPI deficits, consistent with findings in *Celsr3^C1906Y/+^* and *Celsr3^S1894Rfs*2/+^* males. In those models, however, PPI deficits were also observed in females. Notably, PPI deficits were reported in males, but not females, in Celsr3^GFP/+^ mice, which are haploinsufficient and show some similar behavioral changes compared to *Celsr3^C1906Y/+^* and *Celsr3^S1894Rfs*2/+^* TD models [49,54]. PPI deficits in Celsr3^R774H/R774H^ males, despite being present only in homozygous mutants, agree with the idea that PPI deficits and changes to sensorimotor gating may be a defining phenotype in TD mouse models [14]. By contrast to males, *Celsr3^R774H/R774H^* females bury more marbles, suggesting changes to perseverative or compulsive-like behaviors. This data may be reflective of human studies which suggest that complex tics can resemble compulsions often performed in a ritualistic manner and may be more common in females with TD versus males [3,58]. Interestingly, *Celsr3* heterozygous KO mice show no changes to marble burying [54], and *Celsr3^C1906Y/+^* females bury slightly fewer marbles [49]. Although extensive loss of CINs can cause repetitive digging [59], we observed no changes in the amount or distribution of CINs in *Celsr3^R774H/R774H^* mice, suggesting that this phenotype is not caused by changes to interneuron density/location. Although *Celsr3^R774H/R774H^* male and female mice may display some distinct behavioral phenotypes, anatomical changes noted above did not appear to skew according to sex, although larger datasets could show differences.

The milder behavioral changes that *Celsr3^R774H/R774H^* mice show contrast with more widespread and pronounced changes seen in our previously described models and may reflect more retainment of protein function. Unlike *Celsr3^C1906Y/+^* and *Celsr3^S1894Rfs*2^*^/+^ TD models, *Celsr3^R774H/R774H^* homozygous mice are viable at birth and Celsr3 protein levels are unchanged [49]. Notably, both S1894Rfs*2 and C1906Y mutations occur in the second laminin G-like repeat while R774H is located within the fifth cadherin repeat. The fact that R774H affects a different functional domain, one of many extracellular cadherin repeats, and does not change overall protein levels suggest it causes partial loss-of-function effects on the protein, which may account for the milder behavioral phenotypes compared with *Celsr3^C1906Y/+^* and *Celsr3^S1894Rfs*2^*^/+^ TD models. Furthermore, the plethora of neuropsychiatric comorbidities and tic severity along the TD spectrum would seem to support the notion of diverging phenotypes between animal models. Since homozygous *Celsr3^R774H/R774H^* mice were viable and heterozygous animals had no behavioral phenotypes, we focused our studies on homozygotes. It is important to note, however, that the human subjects are all heterozygous for their respective de novo mutations. Furthermore, it is possible that *Celsr3^R774H^* results in haploinsufficiency in humans--but not in mice--which would account for the lack of behavioral phenotypes in heterozygous mutants.

Anatomically, we observed no gross disruptions to major forebrain white matter tracts or changes to cortical layering. Constitutive loss of *Celsr3* affects axon guidance and the development of white matter tracts within the internal capsule, which includes the corticostriatal, corticothalamic, and thalamocortical fibers that comprise CSTC pathways [44,47]. The effects on axon guidance are quite severe and easily observed using lipophillic dyes and widefield fluorescent microscopy. While *Celsr3* is required cell autonomously for corticospinal and corticostriatal axon pathfinding, it guides thalamocortical and corticothalamic axons in a non-cell autonomous manner via its activity in guidepost neurons [44]. Additionally, Celsr3 is required for the formation of axon tracts within basal ganglia circuits [45]. Unlike *Celsr3* constitutive null animals, the *Celsr3^R774H^* amino acid substitution does not cause appreciable misrouting of axons or loss of white matter tracts. However, the ability of axons to terminally branch and/or synapse appropriately onto neurons may be altered, which was not assessed in the present study. Further investigation across multiple *Celsr3* TD models is necessary to determine if such disruptions occur and what their functional impacts might be.

Deep-layer cortical pyramidal neurons in homozygous *Celsr3^R774H^* mice have atrophic basal dendrites, suggesting human mutations in *CELSR3* may affect the ability of neurons to pattern their dendritic arborizations and receptive fields. This unexpected finding hinged upon a leaky germline PV-2A-Cre, which has been previously reported [57]. Reduced complexity of basal dendrites in deep layer pyramidal neurons agrees with previous findings in conditional *Celsr3:Dlx5/6-Cre:Celsr3^FLX/FLX^:Thy1-YFP* KO mice, in which the number and length of basal dendrites on deep layer cortical pyramidal neurons is significantly reduced [56]. Reduced numbers of dendritic spines are also observed. Furthermore, hippocampal CA1 neurons also showed atrophic basal dendrites and loss of dendritic spines in *Celsr3^FLX/FLX^:Foxg1-Cre* mice [55]. Notably, loss of deep layer cortical neurons is also observed in these mutants, presumably due to defects in axon guidance and the failure of subcortical fiber projections to develop, resulting in neuronal cell death [56]. These phenotypes are not observed in *Celsr3^R774H/R774H^* mice, suggesting partial loss-of-function mutations that affect the extracellular cadherin repeats of Celsr3 are still sufficient to alter dendritic patterning. These changes to grey matter are likely functionally important. A more compact dendritic tree limits the temporal and spatial window for integrating coincident excitatory inputs, effectively sharpening the neuron’s tuning to afferent activity [60]. While this increases selectivity for specific sensorimotor inputs, it likely leads to reduced integrative capacity compromising the fidelity of thalamocortical transmission and subsequent cortical output [61]. Such a bias could alter context-dependent modulation of behavior and contribute to the rigid, stimulus-bound action pattern characteristic of TD. Notably, Celsr3 is widely expressed in cortical inhibitory interneurons, raising the possibility that altered pyramidal cell structure interacts with disrupted inhibitory tone to further perturb local circuit computations and downstream read-out [62,63,64].

We also found changes to the types and distributions of dendritic spines along the secondary basal dendrite of cortical pyramidal neurons, with a significant loss of stubby spines and a trend towards an increase in long-thin spines, suggesting the extracellular cadherin repeats of Celsr3 are important both for dendritic patterning and regulating the type/distribution of dendritic spines. We also saw changes to dendritic patterning and reduced spine density along the secondary dendrites of striatal CINs. These morphological changes are consistent with heightened spine turnover and increased synaptic plasticity [65]. Although the functional significance of these changes needs to be tested, this shift in spine distribution may indicate an imbalance between synaptic stabilization and remodeling, affecting how sensorimotor experiences refine cortical circuits over time [66,67]. Taken together, these dendritic and spine-level alterations may constrain the ability of cortical pyramidal neurons (or CINs) to adaptively encode sensorimotor contingencies, which could in turn impact the efficiency and flexibility of corticostriatal communication and control over motor responses. Future studies should investigate the functional consequences of these structural changes in both thalamocortical and corticostriatal pathways, and their behavioral correlates, to better elucidate the circuit-level underpinnings of impaired action control in TD.

*Celsr3* is expressed by ~E13.5 in the mouse ganglionic eminences, which produce cortical and striatal interneurons [68]. While the role of *Celsr3* in the tangential migration of interneurons from the preganglionic eminences has been debated [55], *Celsr3^GFP^* knock-in mice show disrupted tangential interneuron migration and cortical interneuron loss [46]. In these mice, cortical interneuron loss occurs when tangentially migrating interneurons become trapped at the boundary between the cortex and the striatum. This is accompanied by an increase in calretinin-expressing interneurons in the striatum, which are abnormally distributed compared to control animals [46]. Post-mortem studies of brains of adults with severe refractory TD report loss of striatal parvalbumin interneurons and CINs [27,28]. However, CINs and parvalbumin interneurons show normal distribution and density in all reported *Celsr3* TD models to date [49]. Additionally, *Hdc* KO, *Ash1l* heterozygotes, and *WWC1^W88C/W88C^* TD mouse models do not show signs of CIN or parvalbumin interneuron loss in the cortex and/or striatum [52,69,70]. Consistent with these findings, we did not observe loss of cortical parvalbumin or somatostatin interneurons, and the numbers/distribution of CINs was normal in homozygous *Celsr3^R774H/R774H^* mice. Thus, our findings, along with those in previously described TD mouse models, suggest that striatal interneuron loss does not constitute a common biomarker of TD, and may be limited to a subset of subjects at the severe end of the TD spectrum.

Although we have not observed CIN loss in any of our *Celsr3* TD models, we discovered that TD-associated mutations may alter the membrane properties and firing activities of neurons, and potentially spine formation and/or turnover. In *Celsr3^R774H/R774H^* mice, CINs fired APs at a modestly higher frequency than their control counterparts, indicating a subtle shift in intrinsic conductances that govern RMP, AP threshold, and discharge dynamics [71]. Elevated firing in *Celsr3^R774H/R774H^* CINs could reflect more nuanced changes in dopaminergic and/or muscarinic M2 acetylcholine receptor intracellular signaling [72]. Whether this phenotype arises directly from striatal circuit alterations or cell-autonomous effects of the mutant protein or, alternatively, indirectly as a homeostatic adaptation to reduced descending cortical input remains an open question. In the latter case, CIN “up-gain” could serve as a compensatory mechanism: CINs may amplify the gain of corticostriatal information flow due to diminished coincident excitatory drive from pruned deep-layer S1 pyramidal neurons. Through nicotinic receptor facilitation, the increased gain would enhance the salience of weak cortical inputs onto medium spiny neurons (MSNs) [73,74].

The downstream consequences of CIN hyperexcitability are broad. Increased tonic acetylcholine levels would enhance corticostriatal release probability and, at least temporarily, increase collateral inhibition between MSNs in a temporally diffuse manner [75]. Since collateral inhibition is asymmetric, this process would preferentially affect the indirect pathway MSNs [75]. Even small changes in intrinsic excitability can influence the coordinated activity of CIN populations, potentially altering their burst-pause modes which normally entrain with dopamine release [76,77]. Over time, the disruption of pause-rebound patterns in CINs could impair dopamine release dynamics and corticostriatal plasticity, ultimately weakening MSN-MSN inhibitory scaffolding, and compromising ensemble selectivity [78,79]. Therefore, changes in spontaneous CIN firing patterns and dopamine signaling in this model should also be investigated in vivo. Cortical afferents to striatal CINs preferentially originate from the associative cortices [80], suggesting that altered excitability in CINs could further bias the integration of higher-order cortical inputs. Such a mechanism is consistent with functional imaging studies in TD [19,81]. Of note, we did not expand our analyses to other cortical regions, which are an avenue for future studies.

In summary, our findings in *Celsr3^R774H/R774H^* mice point to subtle but detectable changes in behavioral signatures associated with TD; morphological differences suggesting alterations in how cortical neurons pattern their receptive fields within CSTC loops; the capabilities of these neurons to regulate the types and distributions of dendritic spines; and subtle changes in striatal CIN excitability and spine density. These results support the idea that human mutations in *CELSR3* may contribute to TD not through gross structural abnormalities or interneuron loss, but by impairing the integration and signaling capacity of cortical neurons and/or interneurons within critical CSTC circuits. However, whether or not these morphological changes extend to genetic models expressing mutations in other functional domains of *Celsr3* remains to be investigated.

There are some notable limitations to this study. Despite some mild phenotypic differences between sexes, sex was not used as a variable in the anatomical and ex vivo electrophysiological studies, and instead we used a mixed sample of mice from either sex. Additionally, while we did not observe gross anatomical changes to forebrain axon tracts, we cannot rule out finer changes to axon branching or termination patterns, which needs to be investigated at higher resolution using confocal microscopy. This is important considering that sparse cell labeling revealed changes to dendritic patterning in deep layer cortical neurons. Furthermore, some of our analyses, including assessment of dendritic patterning, could benefit from a larger number of samples. However, in many respects, the present findings, including loss of basal dendrite complexity on deep layer cortical neurons and changes to spine density, agree with previous findings in conditional Celsr3 mutant mice [56,57], adding confidence that the results will hold true with larger numbers of mice.

## 4. Materials and Methods

### 4.1. Mouse Lines

All experimental procedures were conducted in accordance with Rutgers Institutional Animal Care and Use Committee (IACUC) guidelines (PROTO201702623, 18 December 2023). Mice were group-housed in individually ventilated cages under a standard 12 h light/dark schedule, with controlled temperature and humidity, and ad libitum access to water and standard chow. Unless otherwise stated, all mice used in this study were young adults (P30–60). Mouse lines used in this study are shown in Appendix A.

CRISPR/Cas9 was used within the Rutgers Gene Editing Shared Resource to produce an R774H amino acid substitution, which maps onto the fifth cadherin repeat (Figure 1a) and corresponds to R783H in the human protein. The following single-stranded oligodeoxynucleotide template was used for targeted insertion via homology directed repair: [CAATCGGCCTGAGTTCACCATGAAAGAGTACCACCTTCGGCTCAATGAGGACGCAGCTGTAGGCACCAGTGTGGTCAGTGTGACTGCGGTAGATCACGATGCTAACAGCGCTATCAGCTACCAAATCACGGGTGGCAACACTCGGAACCGATTTGCCATC]. The following guide RNA was co-injected: [GGTAGTCGATGGTTTAGTGCCCA]. The targeted insertion added a restriction fragment length polymorphism that ablated a site recognized by Taq1 and 15 base pairs downstream of the targeted insertion. Chimeric mice were crossed with wild-type C57BL/6 animals and resulting heterozygous R774H mutant mice were backcrossed again with wild type C57BL/6 mice (Appendix A) for at least three generations. The following Cre recombinase (*Drd1-Cre*, *A2a-Cre*, *Sst-Cre*, *Pvalb-Cre*) and reporter lines (*Celsr3-eGFP*, *Ai14*, *Chat*-eGFP, and *Pvalb*-tdT) were crossed with the *Celsr3^R774H^* line to generate double and triple transgenic lines. The *Celsr3-eGFP* knock-in mouse line was generously provided by Prof. Mario Capecchi, University of Utah, and Prof. Qiang Wu, Shanghai Jiao Tong University [46].

### 4.2. Histology & Immunostaining

#### 4.2.1. Tissue Collection

For the experiments involving neonates, P0 mouse pups (*Celsr3^+/+^* and *Celsr3^R774H/R774H^* littermates, both sexes) were sacrificed by rapid decapitation and brains were quickly removed and processed further. For all other anatomy experiments, young adult (2–4 mo) mice were deeply anaesthetized via intraperitoneal injection of ketamine and xylazine prior to transcardial perfusion with 0.1 M PBS followed by 4% PFA in 0.1 M PBS. Extracted brains were post-fixed in 4% PFA overnight at 4 °C. Following post-fixing, brains were either embedded in 3% agarose and sectioned on a vibratome (Leica VT1200S, Wetzlar, Germany) or incubated overnight in 30% sucrose/0.1 M PBS solution at 4 °C for cryoprotection and sectioned on a cryostat (Leica CM1950, Leica Biosystems, Wetzlar, Germany) the following day.

#### 4.2.2. Immunofluorescent Labeling

The typical protocol for immunofluorescent labelling consisted of 0.1 M PBS washes, followed by a 1–3-h of incubation at room temperature in either normal donkey serum or normal goat serum, depending on the host species of secondary antibodies used. This was followed by incubation in primary antibody solution, at 4 °C overnight, 3–5 washes in 0.1 M PBS, incubation in secondary antibody solution for 1–2 h at room temperature, 3–5 washes in 0.1 M PBS, and finally mounting onto glass microscope slides (VWR, Radnor, PA, USA) using Fluoromount-G mounting media (Southern Biotech, Birmingham, AL, USA). Primary antibodies and concentrations used for fluorescent imaging were as follows: rat anti-L1 (1:500, MAB5272, Millipore, Burlington, MA, USA), NeuroTrace 435/455 Nissl (1:500, N21479, Invitrogen), rabbit anti-µOR (1:1000, 24216, immunoStar, Hudson, WI, USA), mouse anti-Satb2 (1:50, AB51502, Abcam), rat anti-Ctip2 (1:1000, AB18465, Abcam), rabbit anti-Foxp2 (1:1000, AB16406, Abcam), goat anti-parvalbumin (PV) (1:1000, PVG-213, Swant, Burgdorf, Switzerland), rabbit anti-RFP (1:1000, 600-401-379, Rockland), chicken anti-GFP (1:500, GFP-1020, Aves Labs, Davis, CA, USA), goat anti-choline acetyltransferase (ChAT) (1:200, AB144P, Millipore-Sigma), and guinea pig anti-parvalbumin (PV) (1:2000, GP72, Swant). Secondary antibodies for fluorescent imaging: goat anti-rat Alexa Fluor 546 (A11010, Thermo Fisher, Waltham, MA, USA), donkey anti-rabbit Alexa Fluor 647 (A31571, Thermo Fisher), goat anti-mouse Alexa Fluor 546 (A11003, Thermo Fisher), goat anti-rat Alexa Fluor 488 (A11006, Thermo Fisher), goat anti-rabbit Alexa Fluor 647 (A21244, Thermo Fisher), donkey anti-goat Alexa Fluor 488 (A11055, Thermo Fisher), donkey anti-rabbit Alexa Fluor 546 (A11081, Thermo Fisher), donkey anti-chicken Alexa Fluor 488 (A78948, Thermo Fisher), donkey anti-goat Alexa Fluor 546 (A11056, Thermo Fisher), and donkey anti-guinea pig Alexa Fluor 647 (706-605-148, JAX, Bar Harbor, ME, USA).

### 4.3. Western Blot Analysis

Whole brain tissue was collected from neonates. The tissue was homogenized using a Dounce homogenizer and lysed in Synper (Thermo Fisher, 87793). Following lysing, the tissue was spun at 5000× *g* for 10 min. The supernatant was collected for Western blot. A 7% Tris Acetate gel was used for SDS-PAGE (Thermo Fisher, EA0358BOX). Samples were treated with reducing agent and 4X loading dye (Thermo Fisher, NP0009). Following incubation at 95 °C/5 min, 25 µg of protein was loaded into each well. Electrophoresis was run at 150 V for 1 h. Transfer was completed on a PVDF membrane (0.2 µm pore size) on ice for 70 V/1.5 h. The membrane was blocked in 5% non-fat milk for 1 h at room temperature. Primary antibody was incubated overnight at 4 °C. Celsr3 antibody (gift from Fadel Tissir [47]) was used at a ratio of 1:300 and Beta-Actin antibody (75377, Neuromab, Davis, CA, USA) was used at a ratio of 1:2500 in 5% non-fat milk. After incubation, the membrane was washed 6X/5 min in 0.1% PBS-Tween-20. Secondary antibodies were incubated for 30 min at room temperature (Rabbit-anti-Guinea Pig-HRP, Invitrogen PA1-28597, and mouse anti-Rabbit-HRP, SC2357, Waltham, MA, USA). ImageJ version 1.54p was used to calculate the area under the curve for each protein band. A ratio of *Celsr3* to the housekeeper Beta-Actin was used to calculate relative protein concentrations. Statistics were created using Student’s *t*-test.

### 4.4. Viral Sparse Cell Labeling

Mice were anesthetized with 1–3% vaporized isoflurane in oxygen (1 L/min) and placed on a stereotaxic frame. *Pvalb-Cre/+*; *Celsr3^R774H/R774H^* animals and *Pvalb-Cre/+*; *Celsr3^+/+^* littermate control animals were injected with a cocktail of 2 adenoviruses (AAV9-TRE-DIO-vCre and AAV9-TRE-vDIO-GFP-tTA) diluted in sterile saline (1:1:18 ratio of AAV9-TRE-DIO-vCre to AAV9-TRE-vDIO-GFP-tTA to 0.9% NaCl) bilaterally into S1 (+/− 1.80 ML, 0.00 AP, −1.75 DV; 500 nL each injection at a rate of 100 nL/min). This sparse labelling system, provided by Dr. Minmin Luo, Tsinghua University, consists of a controller vector that contains a Tetracycline Response Element promoter (TRE) and a Cre-dependent expressioncassette (double-floxed inverse open reading frame) encoding a mutated Cre-recombinase (vCre) that only recognizes vLoxP sites [57]. The amplifier vector contains a vCre-dependent expression cassette encoding membrane-anchored GFP (mGFP) and the tetracycline-controlled transactivator (tTA) downstream of an internal ribosome entry site. When these viruses are injected into mice that express Cre-recombinase, vCre is flipped into the correct reading frame. vCre can then flip the amplifier expression cassette into the correct orientation, resulting in GFP and tTA expression. Under basal conditions, the TRE promoter is “leaky” and provides very low levels of vCre expression, and only a few neurons will produce enough vCre to flip the amplifier expression cassette into the right orientation. In these sparsely populated neurons, tTA can bind to the TRE promotor on both the control and amplifier vectors, boosting mGFP expression in a positive feedback loop. Following surgery, buprenorphine SR (1.5 mg/kg), carprofen (5 mg/kg) and sterile saline were administered for 3 days post-surgery and the health and welfare of mice were closely monitored. 3 weeks post-surgery, mice were transcardially perfused as described above.

### 4.5. Microscopy & Image Analysis

All anatomy data was acquired using confocal microscopy (Zeiss LSM 700 or Zeiss LSM 800) except for direct and indirect pathway visualization studies where data were collected on a Leica M165FC stereomicroscope with CoolLED illumination. Side-by-side qualitative comparisons were made for—major axon tracts (neonates, 110 µm thick sections on a vibratome), direct and indirect pathways (*Drd1-Cre/+*; *Celsr3^+/+^*; Ai14/+, *Drd1-Cre/+*; *Celsr3^R774H/R774H^*; *Ai14/+*, *A2a-Cre/+*; *Celsr3^+/+^*; *Ai14/+*, and *A2a-Cre/+*; *Celsr3^R774H/R774H^*; *Ai14/+* mice, 120 µm thick sections on a vibratome), and Nissl-B and mu-opioid receptors (*Celsr3^+/+^* and *Celsr3^R774H/R774H^* mice, 40 µm, cryosectioning) using either 10× or 20× objectives and a z-stack tile approach. Quantitative image analysis was done with Imaris (RRID:SCR_007370), Fiji (ImageJ, U.S. National Institutes of Health, Bethesda, MD, US, version 1.54p), and GraphPad Prism 9 (as described below). All images were optimized for presentation using linear adjustments in Fiji (ImageJ). 

#### 4.5.1. Cortical Layer Labeling

Tissue was cryosectioned at 60 µm (*Celsr3^+/+^* and *Celsr3^R774H/R774H^*, both sexes). Images were acquired with a 20× objective and z-stack tile approach. Images were analyzed offline in Imaris. Total cortical depth was measured in S1 cortex from the pial surface to the outer edge of the external capsule. Cortical layer thicknesses were measured along the same axis, guided by the fluorescent layer markers. Cortical layer thicknesses were calculated as a % of total cortical thickness. The Spots function was used within ROIs to determine the density and nearest neighbor distribution of labelled populations within each defined cortical layer. Spots data were exported into Excel (Microsoft) for further analysis.

#### 4.5.2. Interneuron Counting

Fixed brain tissue (Celsr3^+/+^, Celsr3^R774H/R774H^, Sst-Cre/+:Celsr3^+/+^:Ai14/+, Sst-Cre/+:Celsr3^R774H/R774H^:Ai14/+, Celsr3^+/+^:Chat-eGFP and Celsr3^R774H/R774H^:Chat-eGFP) was sliced with a vibratome at 60 µm (for PVIN and SSTIN counts) or 120 µm (for CIN counts) Image data were acquired using a 20× objective and z-stack tile approach with a maximum step size of 2 µm. Images were analyzed offline and blinded to genotype in Fiji (Image J). Interneuron counts were quantitatively compared at 4 predefined anterio-posterior axis positions relative to bregma: position 1 (1.53 to 0.85 mm), position 2 (0.85 to 0.13 mm), position 3 (0.13 to −0.59 mm), and position 4 (−0.59 to −1.31 mm) [73]. 

#### 4.5.3. Anatomical Recovery of Cortical Pyramidal Neurons

Fixed brain tissue was sectioned on a vibratome at 110 µm. mGFP expressing cells were imaged at 20× using a z-stack tile approach with maximum z-steps of 1 µm. For spine counts, secondary dendrites were imaged on a Leica LSM800 confocal microscope using a 63× oil immersion lens with minimum z-step distance (0.46 µm) and post hoc deconvolution. z-stack tile images were imported into Imaris and neurites were semi-automatically traced using the autodepth feature in Filaments. Tracing was performed independently by two different experimenters and blinded to mouse genotype. Somas were rendered using Surfaces for illustration purposes only. Spines were detected semiautomatically, and diameters were recomputed using the shortest distance from distance map algorithm. Spines were classified into 4 distinguished classes: stubby, mushroom, long thin, and filopodia using the ClassifySpines Xtension and specified criteria (Appendix A). All Filaments and ClassifySpines data were exported into Excel for further analysis.

#### 4.5.4. Celsr3 and Interneuron Colocalization

Fixed brain tissue was cryosectioned at 50 µm. Matched striatal slices were mounted and imaged with a 20× objective and a z-stack tile approach with a maximum step size of 3.5 µm. Cell counts were performed using ImageJ ROI manager and manual cell counter.

### 4.6. Behavioral Assays

#### 4.6.1. Prepulse Inhibition of the Acoustic Startle Reflex

Prepulse inhibition was run as previously described on 10–12 week old male and female mice [49,53]. Briefly, mice were placed into a startle chamber above an accelerometer (SR-Lab, San Diego Systems, CA, USA). After a 5 min habituation period mice were subjected to five types of trial: 120 dB startle pulse alone, no pulse, and three prepulse trial types (6, 12 and 16 dB above background) followed by a 120 dB startle stimulus. The intertrial interval averaged 15 s ranging from 8–23 s. The prepulse stimulus was 20ms in length and the startle pulse was 40ms in length. Background white noise was set to 65 dB.

#### 4.6.2. Open Field Arena

Mice (*Celsr3^+/+^* and *Celsr3^R774H/R774H^*, both sexes) were brought to the experiment room and allowed to habituate for 30 min. Individual mice were placed at the same edge of the open field arena (40 cm × 40 cm, Med Associates, legacy, Fairfax, VT, USA) and allowed to freely explore for 30 min. The center of the arena was defined as the middle 10 cm × 10 cm of the arena.

#### 4.6.3. Accelerated Rotarod Test

The Rota-rod apparatus (LE8205, Panlab, Hollistin, MA, USA) was used to assess motor learning capabilities of mice. Mice were placed on the rod and the rod was started at 4 rpm and accelerated to 40 rpm in a timespan of 5 min. The time each mouse fell off the rotarod was recorded automatically (latency). Mice were given at least 1 min recovery time between trials. Mice were tested for 5 trials per day over 6 consecutive days. The apparatus was cleaned and dried between trials. The average latency to fall for each mouse per day was plotted.

#### 4.6.4. Marble Burying Assay

Mice (*Celsr3^+/+^* and *Celsr3^R774H/R774H^*, both sexes) were gently placed into a rectangular arena with a 5 cm base of Beta Chip bedding (Northeastern Products, Warrensburg, NY, USA), with 20 glass marbles placed on top of the bedding in a 4 × 5 matrix. After 30 min, each mouse was returned to its home cage and the number of marbles buried counted. Marbles were counted as ‘buried’ if 50% or more was underneath the bedding.

### 4.7. Ex Vivo Electrophysiology

Mice (*Celsr3^+/+^*; *Chat-eGFP* and *Celsr3^R774H/R774H^*; *Chat-eGFP*, both sexes) were anesthetized with an intraperitoneal injection of ketamine-xylazine solution prior to rapid decapitation and brain dissection [82]. Coronal 300 µm sections were taken on a vibratome (Leica VT1200S) in ice-cold sucrose substituted cerebrospinal fluid (aCSF) containing (in mM): 250 sucrose, 25 NaHCO_3_, 10 glucose, 2.5 KCl, 1 NaH_2_PO_4_, 1 MgCl and 2.5 CaCl_2_. Ringers’ solutions were continually bubbled with 95% O_2/_5% CO_2_ to maintain oxygenation and neutral pH. Sections were allowed to recover for 1 h at room temperature in normal aCSF (118 mM NaCl substituted for sucrose) prior to recording. aCSF was continually bubbled with 95% O_2/_5% CO_2_. Evoked action potential characterization was done using a potassium gluconate based internal solution containing (in mM): 135 K.gluconate, 8 NaCl, 10 HEPES, 0.1 EGTA, 0.3 Na_3_GTP, and 2 Mg_2_ATP. Biotin hydrobromide (0.2%, Biotium) was added to the internal solution. Data were amplified using a Multiclamp 200B amplifier, digitized using a Digidata 1550A, and acquired using pClamp11 (Molecular Devices, CA, USA, RRID:SCR_011323). Series resistance (R_s_), membrane resistance (R_m_), membrane capacitance (C_m_), and resting membrane potential (RMP) were measured at the beginning of recording and monitored throughout. Evoked AP characteristics were recorded within 1 min of membrane breakthrough. Bridge balances were applied in current clamp mode. Voltages have not been corrected for liquid junction potential. Cholinergic interneurons in the dorsolateral striatum were fluorescence targeted via their expression of eGFP, and their identities were confirmed physiologically via relatively depolarized RMPs (~−55mV), prominent voltage sag, slow AHP currents, and relatively wide action potential waveforms. AP threshold was measured using the first derivative of the AP and was defined as the voltage at which dV/dt = 10 mV·s^−1^. Data were excluded from analysis if R_s_ > 30 MOhm or if ∆Rs > 20% over the course of the recording. Electrophysiology data were analyzed offline in Axograph X (Axograph, Sydney, Australia, RRID:SCR_014284). At the end of recording, slices were dropped into 4% PFA for post hoc anatomical recovery. Slices were kept in 4% PFA overnight at 4 °C, washed in 0.1 M PBS, then stored in 0.1 M PBS + 5 mM NaN_3_ at 4 °C until further processing.

### 4.8. Statistical Analyses

Parametric methods were used whenever possible to compare between two or more means. Normality was assessed using Shapiro–Wilk tests, and Levene’s test was used to assess homogeneity of variances. Main and interaction effects were evaluated for significance using an α level of 0.05. All comparisons were made with two-tailed tests, unless otherwise specified. Significant effects were followed by pre-planned multiple comparisons, with Bonferroni corrected α values. Unless otherwise specified, data was presented as mean ± standard error of the mean (SEM). For data that would not satisfy parametric test assumptions, non-parametric alternatives were used, and data is shown as median ± inter quartile range (IQR). All descriptive and inferential statistical analyses were done using GraphPad Prism 9 (RRID:SCR_002798; GraphPad Software, Boston, Massachusetts USA, http://www.graphpad.com (accessed on 1 June 2021)).

## Figures and Tables

**Figure 1 ijms-26-10307-f001:**
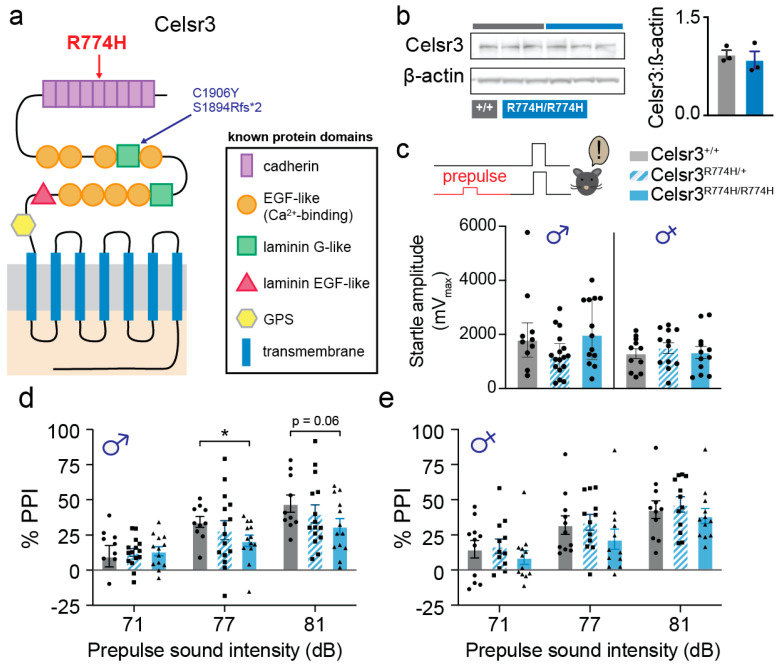
R774H mutation causes no changes to relative Celsr3 protein levels, and *Celsr3^R774H^*-mutant mice show mild PPI deficits. (**a**) Known domains of *CELSR3* protein and location of arginine to histidine substitution (R774H, red arrow) within the fifth cadherin repeat, as well as previous mutations that were tested in mice that are located in laminin G-like domain (C1906Y, and S1894Rfs*2, red arrow). (**b**) Western blot results for *Celsr3^+/+^* and *Celsr3^R774H/R774H^*. Relative protein levels that were obtained from the gel (**left**), are plotted on the right. Protein levels are comparable between genotypes, indicating preserved Celsr3 amounts (*t*(4) = 0.54, *p* = 0.62, independent samples *t*-test). (**c**) Prepulse inhibition (PPI) paradigm, mice did not show any difference in baseline startle magnitudes (left: males (presented as median ± IQR), Krusakal Wallis, *p* = 0.072 (approximate); right: females, F(2, 32) = 0.2761, *p* = 0.76, ordinary 1-way ANOVA). (**d**) Male homozygous mice (*Celsr3*^R774H/R774H^, triangles) had significantly reduced PPI compared to heterozygous (*Celsr3*^R774H/+^, squares) and wild-type littermate controls (*Celsr3*^+/+^, circles; F(1, 21) = 4.675, *p* = 0.042, 2-way ANOVA, Sidak’s multiple comparisons testing, pp71: *p* = 0.6, pp77: *p* = 0.03 and pp81, *p* = 0.06). (**e**) Female mice showed no significant changes to their PPI phenotype (F(1, 21) = 0.7, *p* = 0.4; *Celsr3*^+/+^/*Celsr3*^R774H/+^/*Celsr3*^R774H/R774H^, males: *n* = 10/15/13, Females: *n* = 11/12/12). *p* < 0.05 (*).

**Figure 2 ijms-26-10307-f002:**
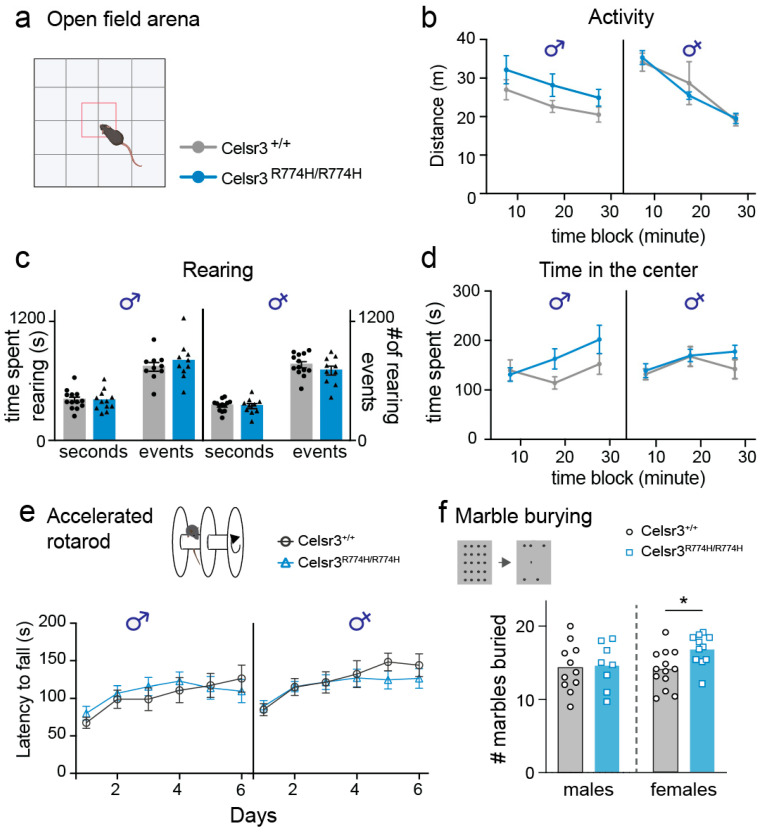
*Celsr3^R774H^*-mutant mice do not exhibit hyperactivity or motor learning deficits, but females show increased compulsive behavior. (**a**) Open field test was used to assess activity levels and exploratory behavior in *Celsr3* mice (*Celsr3^+/+^*/*Celsr3^R774H/R774H^*, males: *n* = 10/12, females: *n* = 12/12) (**b**) Open field results show no changes to distance traveled in the open field, indicating no hyperactivity (2-way RM-ANOVA, males: F(1,20) = 2, *p* = 0.17, females: F(1,21) = 0.02, *p* = 0.88). (**c**) Total rearing time (Student’s *t*-test, males: *p* = 0.87, females: *p* = 0.87) and events (Student’s *t*-test, males: *p* = 0.45, females: *p* = 0.28) were similar between genotypes (*Celsr3^+/+^*: circles, *Celsr3^R774H/R774H^*: triangles). (**d**) Time spent in the center region of the open field arena was comparable between genotypes (2-way RM-ANOVA, males: F(1.20) = 1.6, *p* = 0.22, and females: F(1,20) = 0.5, *p* = 0.45); however, *Celsr3^R774H/R774H^* mice of both sexes spent significantly more time in the center in the last 10 min of testing compared to the first 10 min (Sidak’s multiple comparisons test, males: *p* = 0.04, females: *p* = 0.049) *Celsr3^+/+^* behaved similarly in the first ten minutes and the last ten minutes of testing (Sidak’s multiple comparisons test, males: *p* = 0.94, females, 0.87). (**e**) Rotarod test in the accelerated condition revealed no significant changes to motor adaptation between genotypes (2-way RM-ANOVA, males: F(1,17) = 0.09, *p* = 0.76, females: F(1,17) = 0.25, *p* = 0.62, *Celsr3^+/+^*/*Celsr3^R774H/R774H^*, males: *n* = 9/10, females: *n* = 11/8). (**f**) Marble burying assay revealed increased perseverative-like behavior in *Celsr3^R774H/R774H^* female mice (Student’s *t*-test, males: *p* = 0.5, females: *p* = 0.013, *Celsr3^+/+^*/*Celsr3^R774H/R774H^* males: *n* = 12/8, females: *n* = 14/11). *p* < 0.05 (*).

**Figure 3 ijms-26-10307-f003:**
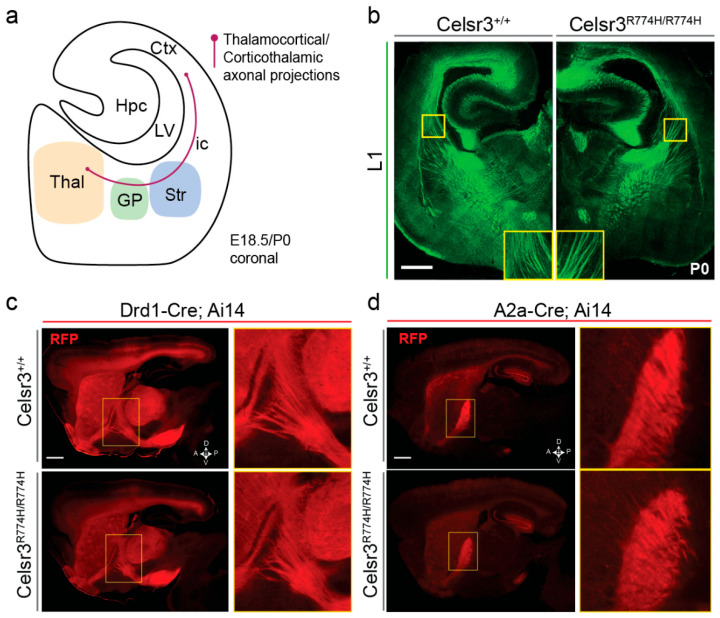
Homozygous point mutation in Celsr3 does not perturb gross organization of the mouse brain. (**a**) Depiction of P0 axonal projections—thalamocortical and corticothalamic projections (coronal view at axial position shown in panel b). Neonate (P0) brains were compared using different markers to reveal gross anatomical changes. (**b**) L1 antibody labelling in coronal sections of the P0 mouse brain shows fiber tracts in *Celsr3^+/+^* (**left**) and *Celsr3^R774H/R774H^* (**right**) mice. Scale bar represents 500 µm. (**c**) Sagittal view of direct pathway axon tracts in adult *Celsr3^+/+^* (**top**) and *Celsr3^R774H/R774H^* (**bottom**) based on *Ai14* expression under control of *Drd1-Cre*. (**yellow rectangle**) zoom in of direct pathway axon tracts (**d**) Sagittal view of indirect pathway fiber tracts in *Celsr3^+/+^* (**top**) and *Celsr3^R774H/R774H^* (**bottom**) mice based on *Ai14* expression under control of *A2a-Cre*, (**yellow rectangle**) zoom in of indirect fiber tracts. Scale bar represents 1 mm. Thal—thalamic nuclei, GP—globus pallidus, Str—striatum, Hpc—hippocampus, LV—lateral ventricle, Ctx—cortex, ic—internal capsule.

**Figure 4 ijms-26-10307-f004:**
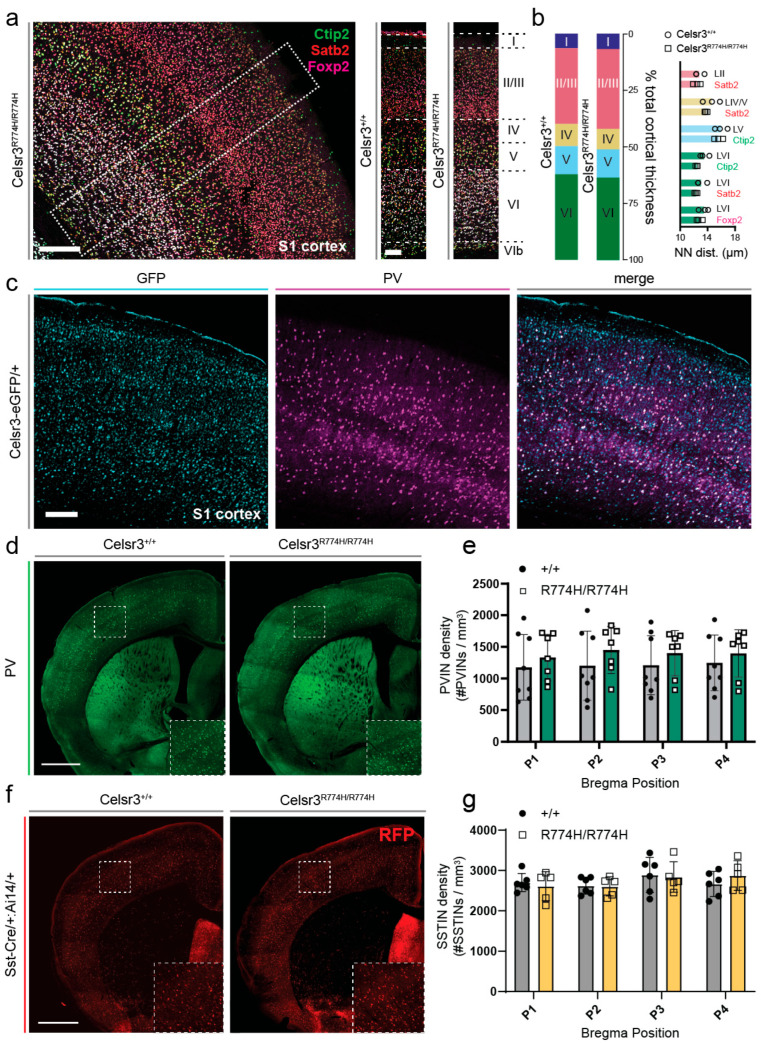
Cortical layering and inhibitory interneuron patterning is not significantly impacted by the R774H amino acid substitution in *Celsr3*. (**a**) Representative image of cortical layers in a *Celsr3^R774H/R774H^* mouse somatosensory (S1) cortex (large image). Scale bar represents 200 µm. Representative ROIs of *Celsr3^+/+^* and *Celsr3^R774H/R774H^* (smaller images), with layer positions I-VI marked. Scale bar represents 100 µm. (**b**) Relative cortical layer thickness in *Celsr3^+/+^* (left bar, *n* = 3) and *Celsr3^R774H/R774H^* (right bar, *n* = 3, *p* = 0.9742, Chi-square test). Nearest neighbor distances across labelled populations within defined layers (*p* = 0.2275, 2-way ANOVA). (white rectangle) zoom in of cortical layers (**c**) *Celsr3-eGFP* expression in mouse S1 cortex co-labelled for parvalbumin (PV). Scale bar represents 200 µm. (**d**) Representative images of cortical PVINs in *Celsr3^+/+^* (**left**) and *Celsr3^R774H/R774H^* (**right**) mice. Scale bar represents 1 mm. (white rectangle) zoom in of cortical PVINs (**e**) Comparison of cortical PVIN density at four different AP positions (*Celsr3^+/+^ n* = 8, *Celsr3^R774H/R774H^ n* = 7, *p* = 0.4159, 2-way ANOVA). (**f**) Representative images of cortical SSTINs in *Sst-Cre/+*:*Celsr3^+/+^*:*Ai14/+* (**left**) and *Sst-Cre/+*:*Celsr3^R774H/R774H^*:*Ai14/+* (**right**) mice. Scale bar represents 1 mm. (white rectangle) zoom in of cortical SSTINs (**g**) Comparison of cortical SSTIN density at four different AP positions (*p* = 0.8944, 2-way ANOVA).

**Figure 5 ijms-26-10307-f005:**
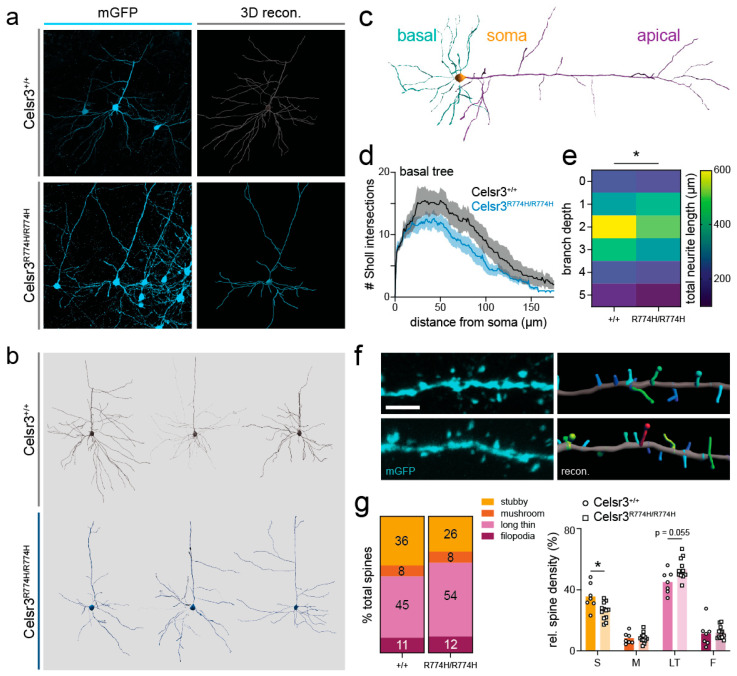
Basal dendrites of *Celsr3*-mutant cortical pyramidal neurons show reduced complexity. (**a**) Representative images of confocal images (**left**) and their 3D reconstructions (**right**). Scale bar represents 100 µm. (**b**) Representative reconstructions of cortical pyramidal neurons from *Celsr3^+/+^* (**top**, grey) and *Celsr3^R774H/R774H^* (**bottom**, blue) mice (*n* = 3 mice for each genotype). Scale bar represents 50 µm. (**c**) Schematic showing denotation of basal dendrites (blue) versus apical dendrites (purple). (**d**) Sholl analysis of basal dendrites of *Celsr3^+/+^* (*n* = 6, black) and *Celsr3^R774H/R774H^* (*n* = 8, blue) neurons (genotype effect *p* < 0.001, 2-way ANOVA). Shaded area represents SEM. (**e**) Heat map comparing total neurite length vs. branch depths (*p* = 0.0271, 2-way ANOVA). (**f**) Representative confocal images of secondary dendrites (**left**) and their 3D reconstruction and classification (**right**). Scale bar represents 2 µm. (**g**) Relative spine density by class: stubby (S), mushroom (M), long thin (LT) and filopodia (F) in *Celsr3^+/+^* and *Celsr3^R774H/R774H^* mice (stubby spines *p* = 0.03, long thin spines *p* = 0.055, multiple Holm-Šídák *t*-test with multiple comparison correction). *p* < 0.05 (*).

**Figure 6 ijms-26-10307-f006:**
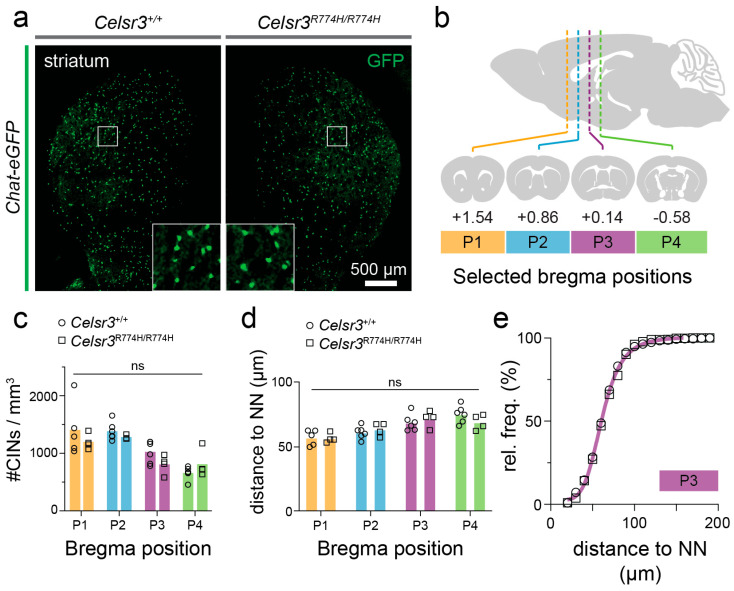
*Celsr3^R774H^*-mutant mice have no detectable loss of cholinergic striatal interneurons. (**a**) Representative images of *Celsr3^+/+^*; *Chat-eGFP* and *Celsr3^R774H/R774H^*; *Chat-eGFP* striatum. Scale bar represents 500 µm. (**b**) Four axial positions were chosen to quantify the density of interneurons. (**c**) Density of GFP+ Cholinergic interneurons in *Celsr3^+/+^* (*n* = 7) and *Celsr3^R774H/R774H^* (*n* = 10) striatum at four axial positions (*p* = 0.6728, 2-way ANOVA, ns: not significant). (**d**) Distance to the nearest neighbor (NN) from counted cholinergic interneurons in *Celsr3^+/+^* (*n* = 7) and *Celsr3^R774H/R774H^* (*n* = 10) striatum at four axial positions (*p* > 0.05, 2-way ANOVA). (**e**) Distribution of distances to cholinergic neurons’ nearest neighbors in one representative axial position (P3).

**Figure 7 ijms-26-10307-f007:**
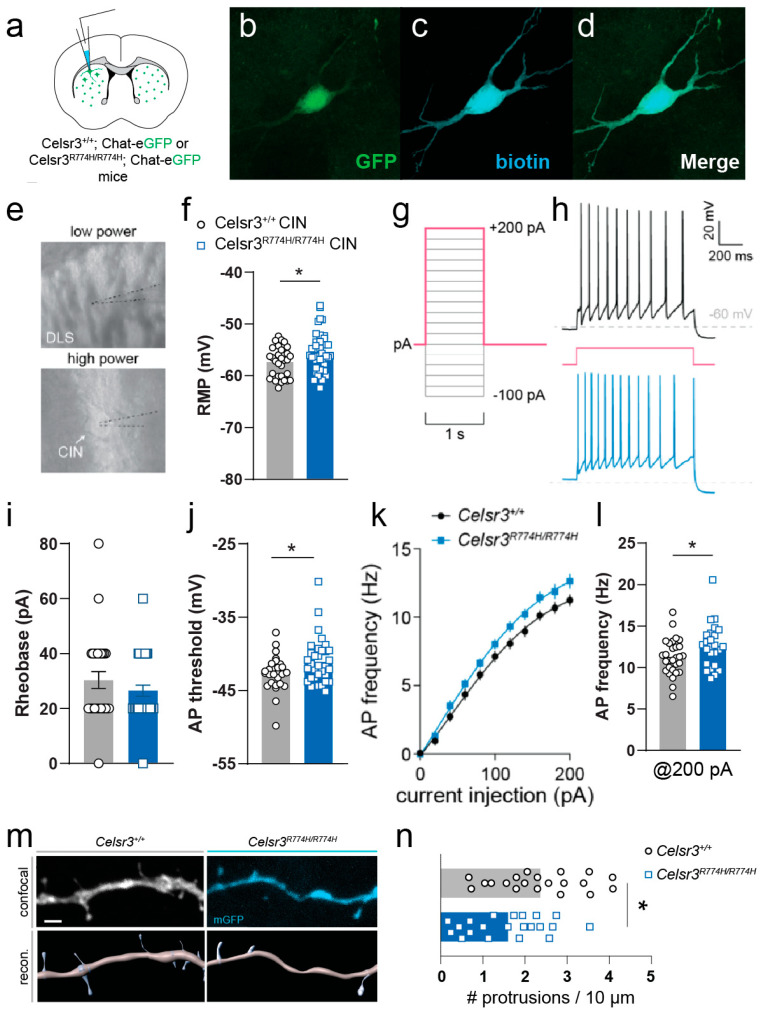
Striatal cholinergic interneurons show mild intrinsic hyperexcitability. (**a**) Schematic of a coronal slice from mouse brain showing approximate AP levels, and locations of recordings. 300 um slices were made from double-transgenic mice (Celsr3^+/+^;ChAT-GFP *n* = 8, or Celsr3^R774H/R774H^; ChAT-GFP *n* = 7) to record (**b**) endogenously GFP expressing cholinergic interneurons (CINs). (**c**) Recorded cells were filled with biotin, (**d**) which were verified to coexpress GFP post hoc. (**e**) DIC images during recording at low power (**top**) showing placement of electrode in the dorsolateral striatum and high power (**bottom**) showing placement of electrode on an identified CIN. (**f**) Resting membrane potential (RMP) of recorded *Celsr3^+/+^* (*n* = 31) and *Celsr3^R774H/R774H^* (*n* = 35) Cholinergic interneurons (*p* = 0.0386, two-tailed *t*-test). (**g**) Depolarizing current injection ladder used to characterize evoked action potentials in current clamp mode. (**h**) Representative traces of a *Celsr3^+/+^* (**top**, black trace) and a *Celsr3^R774H/R774H^* (**lower**, blue trace) tonically firing CIN in response to 200 pA current injection (red step). (**i**) Rheobase of *Celsr3^+/+^* and *Celsr3^R774H/R774H^* Cholinergic interneurons (*p* = 0.3505, Mann–Whitney test). (**j**) Action potential (AP) threshold of *Celsr3^+/+^* and *Celsr3^R774H/R774H^* Cholinergic interneurons (*p* = 0.0456, unpaired *t*-test). (**k**) f/I plot of *Celsr3^+/+^* (*n* = 26) and *Celsr3^R774H/R774H^* Cholinergic interneurons (*n* = 19, **left plot**, *p* < 0.0001, nonlinear fit—different curve for each dataset). (**l**) AP frequency @ 200 pA injection for *Celsr3^+/+^* and *Celsr3^R774H/R774H^* Cholinergic interneurons (**right graph**, *p* = 0.0382, two-tailed t test). (**m**) Representative images of confocal maximum intensity projections of second order dendrite ROIs in *Celsr3^+/+^* (**left**, **top**) and *Celsr3^R774H/R774H^* (**right**, **top**) mice and their 3D reconstructions and semiautomatic spine detection (lower panels). Scale bar represents 2 µm. (**n**) Average spine density on second order dendrites (*p* = 0.0184, *t*-test). *p* < 0.05 (*).

## Data Availability

The data is available from the corresponding author upon request.

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
