# Peer review of "A Celsr3 Mutation Linked to Tourette Disorder Disrupts Cortical Dendritic Patterning and Striatal Cholinergic Interneuron Excitability"

_ijms, 2025, doi:10.3390/ijms262110307_

Round 1

Reviewer 1 Report

Comments and Suggestions for Authors

The authors establish a viable animal model carrying a human disease-associated variant and provide behavioral, anatomical, and electrophysiological phenotypes of TD. This study shows a translational potential for investigating clinically oriented abnormalities in a subtype of TD. However, the data processing and analytical methods require further refinement. Here are some comments on each experiment and section below, organized point by point.

  1. Because cholinergic interneurons are scarce in the cortex, a cortical interpretation is unlikely. However, the title would be clearer as “striatal cholinergic interneurons.”
  2. Since the authors have deep expertise in Celsr3, their coverage in the Introduction and Discussion is particularly detailed and insightful.
  3. Because parametric methods are reported for all figures in this study (which implicitly assume normality in each group), it is necessary to provide a Statistical Analysis paragraph indicating that normality was verified for each group and that analyses were selected in accordance with the underlying distributions. On the other hand, datasets that are non-normally distributed should be analyzed with non-parametric methods and displayed as median ± IQR rather than mean ± SEM.
  4. Please provide all data points in the figure, including 4e, 4g.
  5. I think the Materials and Methods are oddly structured. Immunohistochemistry should be presented in a single subsection and the various quantification procedures in another, rather than interleaving staining and quantification by experiment. Many IHC quantification steps are insufficiently described, leaving it unclear how the reported numbers were obtained.
  6. Because the manuscript highlights gender-specific behavioral differences (especially the Discussion emphasizes a gender-specific effect), all subsequent experiments must indicate the gender analyzed. At present, aside from the interneuron colocalization experiment (line 719, males), the sex of subjects is not specified, making it difficult to connect behavior with underlying anatomical and physiological deficits.
  7. 3b, c. Axon projections should be quantified rather than described qualitatively. Given the thick sections and confocal z-stacks, volumetric comparisons (e.g., axon volume fraction) are feasible and would strengthen the claims.
  8. 3c, d. The current resolution of images is insufficient to assess direct vs. indirect pathway projections or striatal compartmentation; please provide higher-resolution panels with zoomed insets. The striatal compartmentation varies from rostral to caudal levels. The ratio of MOR1-positive and -negative regions can be quantified along anatomical levels as well.
  9. 4a, b. While the mapping of TBR1/CTIP2/SATB2 to cortical layers may be common knowledge for experts, it should be stated explicitly. That being said, in Fig. 4b, some layer designations are difficult to interpret, e.g., FOXP2 is not a canonical layer II/III marker. The rationale for quantifying “layer II/III FOXP2” needs to be explained or revised.
  10. 4a. The Magenta/Red/Green palette is hard to distinguish; please switch to Cyan/Magenta/Yellow or a standard RGB scheme.
  11. 4d, f. Because interneuron density varies systematically across cortical layers and Celsr3 regulates interneuron migration, quantification should be layer-resolved (at least in the sensorimotor cortex). Accordingly, Fig. 4d, f should present laminar counts (e.g., L2/3, L4, L5, L6) normalized to layer area/volume rather than from a single S1 ROI.
  12. 4c-f. Given access to Imaris, it should be feasible to report the interneuron/reporter colocalization rate and interneuron density.
  13. 5. The cellular sample size is small, and the number of mice analyzed is not reported; thus, it is impossible to evaluate how much the result is driven by between-subject differences.
  14. 7a. Two animals per group is insufficient for inferential statistics; please increase the number of animals before drawing conclusions.
  15. The Discussion would benefit from an initial paragraph that provides a high-level synthesis and frames the contribution, enabling readers to link the individual results into a coherent narrative.
  16. While you note that it remains unclear whether CIN changes are direct or indirect in the Discussion, the authors should provide some explanation or hypothesis of mechanisms linking the S1 pyramidal phenotype (reduced spine density/branching) to altered CIN excitability, e.g., direct corticostriatal synapses onto CINs (10.7554/eLife.35657) and indirect modulation via SPN collateral inhibition (10.1523/JNEUROSCI.3833-10.2011; 1523/JNEUROSCI.5493-07.2008).

Author Response

Revisions Round 1 – 10/3/25 

We thank the reviewers and editors for their thoughtful and constructive feedback on our manuscript, “A cadherin mutation in Celsr3 linked to Tourette Disorder affects dendritic patterning and excitability of striatal cholinergic interneurons”. 

We have carefully revised the manuscript in accordance with the reviewers’ comments, ensuring that the revised portions are clearly highlighted, and suggested literature has been evaluated. Where appropriate, we have incorporated the reviewers’ recommendations, and in cases where a comment could not be addressed directly, we provide a detailed explanation in our responses below. We believe these revisions have substantially strengthened the clarity and impact of the manuscript, and we are grateful for the opportunity to improve our work. 

(All text changes based on reviewer comments are highlighted in yellow in the updated manuscript) 

Comments: 

Reviewer #1: 

  1. Because cholinergic interneurons are scarce in the cortex, a cortical interpretation is unlikely. However, the title would be clearer as “striatal cholinergic interneurons.” –  

Response: We added “striatal” to the title in line 3. 

  1. Since the authors have deep expertise in Celsr3, their coverage in the Introduction and Discussion is particularly detailed and insightful. 

Response: Thank you for the positive feedback.  

  1. Because parametric methods are reported for all figures in this study (which implicitly assume normality in each group), it is necessary to provide a Statistical Analysis paragraph indicating that normality was verified for each group and that analyses were selected in accordance with the underlying distributions. On the other hand, datasets that are non-normally distributed should be analyzed with non-parametric methods and displayed as median ± IQR rather than mean ± SEM.  

Response: We have added a statistical analysis paragraph in the methods section (see lines 896-906).  When re-checking for normality, we discovered that the male startle response (figure 1c) was not normally distributed. We therefore changed our analysis from a parametric one-way ANOVA to nonparametric Kruskal Wallis test. The figure has been changed to show median/IQR instead of Mean/SEM. The new statistics are incorporated into the legend for figure 1 in lines 157-159. 

  1. Please provide all data points in the figure, including 4e, 4g.  

Response: We added the individual data points to each graph in panels 4e and 4g. We note that the bimodal distributions seen for both wild-type and mutant cell count reflects variability in PV interneuron staining/intensity that we often encounter in thick tissue sections. Although we did not include this data, we see similar variability when using PV-tdTomato reporter mice in both mutant and wild-type animals.   

  1. I think the Materials and Methods are oddly structured. Immunohistochemistry should be presented in a single subsection and the various quantification procedures in another, rather than interleaving staining and quantification by experiment. Many IHC quantification steps are insufficiently described, leaving it unclear how the reported numbers were obtained.  

Response: Thank you, we have reorganized the materials and methods accordingly. 

  1. Because the manuscript highlights gender-specific behavioral differences (especially the Discussion emphasizes a gender-specific effect), all subsequent experiments must indicate the gender analyzed. At present, aside from the interneuron colocalization experiment (line 719, males), the sex of subjects is not specified, making it difficult to connect behavior with underlying anatomical and physiological deficits.  

Response: We used a mixed gender cohort. Our previous studies found no changes in density or distribution when we looked at interneurons in mice with a stronger phenotype than the model presented in this paper. Reference: (https://doi.org/10.1073/pnas.230715612)  

In addition, we do not want to overstate the apparent sex differences, because they are mainly limited to a couple of different behavioral phenotypes. In lines 512-514 we now write: 

Although Celsr3R774H/R774H male and female mice may display some distinct behavioral phenotypes, anatomical changes noted above did not appear to skew according to sex, although larger datasets could show differences.   

  1. 3b, c. Axon projections should be quantified rather than described qualitatively. Given the thick sections and confocal z-stacks, volumetric comparisons (e.g., axon volume fraction) are feasible and would strengthen the claims.  

Response: We used the word “qualitative,” because we do not have high-resolution confocal images in Z stacks. Rather, these images were acquired from thick tissue sections using widefield fluorescent stereomicroscopy. We now include magnified regions of interest to show fasciculated axons without signs of probst bundles or wandering axons. Thus, from a gross organizational level, we do not see major changes to axon tracts that have been reported in Celsr3 null mice using widefield fluorescent microscopy, or conditional KO models. This further speaks to the nature of the mutation, likely causing partial loss-of-function effects on the protein. We also discuss this in the final paragraph of the discussion, which notes certain limitations in the current study. Specifically, we write, “Additionally, while we did not observe gross anatomical changes to forebrain axon tracts, we cannot rule out finer changes to axon branching or termination patterns, which needs to be investigated at higher resolution using confocal microscopy. This is important considering that sparse cell labeling revealed changes to dendritic patterning in deep layer cortical neurons.” in lines, 573-577. 

  1. 3c, d. The current resolution of images is insufficient to assess direct vs. indirect pathway projections or striatal compartmentation; please provide higher-resolution panels with zoomed insets. The striatal compartmentation varies from rostral to caudal levels. The ratio of MOR1-positive and -negative regions can be quantified along anatomical levels as well.  

Response: We have provided magnified insets that show axons terminating in the GPe/GPi. From the resolution afforded by widefield microscopy, we do not see any major differences. Of note, although we did not include this data, Celsr3 R774H/R774H mice do not show any noticeable differences to descending cortical projections, as determined using confocal microscopy and  Emx1-Cre:Ai14 to label axon terminals. Given the various approaches, we decided not to pursue more in-depth studies at that time, although we initially hypothesized that axon guidance would be affected in this model (and to our great disappointment when we did not observe noticeable changes). We also decided to omit the MOR1 staining panels as we could not locate several of the original source images.  

  1. 4a, b. While the mapping of TBR1/CTIP2/SATB2 to cortical layers may be common knowledge for experts, it should be stated explicitly. That being said, in Fig. 4b, some layer designations are difficult to interpret, e.g., FOXP2 is not a canonical layer II/III marker. The rationale for quantifying “layer II/III FOXP2” needs to be explained or revised. 

Response: Thank you for this observation and suggestion. We commonly see ‘background’ FoxP2 labeling in cortical layers II/III but, upon further reflection, we agree that this marker is not suitable for demarcating layer II/III neurons. We are omitting FOXP2 quantification from layer II in our nearest neighbor analysis (fig. 4B). However, the overall take home message remains the same, as our data does not suggest major changes to cortical layering (consistent with findings in Celsr3 null mice), and the nearest neighbor analysis does not suggest neuron loss or changes to radial neuron migration (even upon omission of FoxP2 staining in layers II/III). Given the canonical expression pattern of FOXP2, we have kept this data for the deeper cortical layers.  

  1. The Magenta/Red/Green palette is hard to distinguish; please switch to Cyan/Magenta/Yellow or a standard RGB scheme. 

Response: We have changed the color scheme to make it clearer.  

  1. 4d, f. Because interneuron density varies systematically across cortical layers and Celsr3 regulates interneuron migration, quantification should be layer-resolved (at least in the sensorimotor cortex). Accordingly, Fig. 4d, f should present laminar counts (e.g., L2/3, L4, L5, L6) normalized to layer area/volume rather than from a single S1 ROI.  

Response: We do not have layer-specific interneuron counts, but we initially assessed changes to deep versus superficial layers (using arbitrary boundaries) and did not see any noticeable differences. Furthermore, we decided against pursuing more rigorous cell counts as our results are consistent with Zhou et. Al 2008 (10.1126/science.1155244) in which tangential interneuron migration is reported as normal in Celsr3 KO mice—despite later reports in a transgenic GFP knock-in mouse that suggested tangential interneuron migration could be affected in embryos.   

4c-f. Given access to Imaris, it should be feasible to report the interneuron/reporter colocalization rate and interneuron density. 

Response: We apologize if we misunderstand the critique, but we did not use IMARIS because nuclear markers provide sufficient resolution for quantifying co-localization between ChAT/PV staining and GFP. Of note, the average distance between cholinergic interneurons was quantified using IMARIS and nearest neighbor analysis, revealing no differences in cell density or distribution.  

  1. The cellular sample size is small, and the number of mice analyzed is not reported; thus, it is impossible to evaluate how much the result is driven by between-subject differences.

We analyzed neurons from 3 different mice of each genotype. Due to the nature of the sparse-cell labeling, the total number of neurons analyzed is somewhat limited. That being said, the differences were striking between mutant and control neurons, despite the limited data set, and agree with previous observations noted in Celsr3 conditional KO mutants that we cite in the discussion.  

  1. Two animals per group is insufficient for inferential statistics; please increase the number of animals before drawing conclusions.  

Response: We analyzed another animal and now have an n=3 for each group. 

  1. The Discussion would benefit from an initial paragraph that provides a high-level synthesis and frames the contribution, enabling readers to link the individual results into a coherent narrative.  

Response: Agreed. We added an initial summary paragraph to the discussion and have restructured the discussion to provide a higher-level synthesis of the results and their relationship to other TD models and Celsr3 KO models in lines 472-496.  

  1. While you note that it remains unclear whether CIN changes are direct or indirect in the Discussion, the authors should provide some explanation or hypothesis of mechanisms linking the S1 pyramidal phenotype (reduced spine density/branching) to altered CIN excitability, e.g., direct corticostriatal synapses onto CINs (10.7554/eLife.35657) and indirect modulation via SPN collateral inhibition (10.1523/JNEUROSCI.3833-10.2011; 1523/JNEUROSCI.5493-07.2008).  

Response: We have incorporated a discussion of a potential mechanism through which the changes to CIN membrane properties might be indirect, as well as more information supporting a more parsimonious explanation (direct changes) in Supplementary figure S2. Please see the expanded discussion which includes your reference suggestions (lines 607-637). 

Reviewer 2 Report

Comments and Suggestions for Authors

This manuscript by Nasello et al. presents a thorough and sophisticated investigation into the effects of a novel Celsr3 R774H point mutation, linked to Tourette Disorder (TD), on brain development and function in a mouse model. The study is commendable for its multi-level approach, combining sophisticated genetics, detailed anatomical analyses, behavioral phenotyping, and electrophysiology. The finding that this mutation in the fifth cadherin repeat leads to subtler, yet functionally significant, changes in dendritic patterning and interneuron excitability—without the gross anatomical defects or lethality associated with mutations in other domains—is a significant contribution to the field. It effectively supports the emerging paradigm that TD-related neuropathology may involve nuanced circuit dysfunction rather than overt cell loss or miswiring. The work is overall well-executed and of high interest. However, several major and minor points require attention to strengthen the manuscript and ensure clarity and reproducibility.

Major Points:

1. The authors focus their core analysis on homozygous Celsr3R774H/R774H mice. However, it is noted that human TD-associated mutations in CELSR3 are heterozygous. While the rationale for using homozygotes (stronger phenotype, viability) is understandable, the relevance of the findings to the human heterozygous condition should be explicitly discussed as a potential limitation of the model. Furthermore, a critical inconsistency must be clarified: the Western blot in Figure 1b is described in the main text as comparing Celsr3+/+ and Celsr3R774H/R774H, yet the figure legend lists Celsr3R774H/+ (heterozygotes). The genotypes used in this crucial experiment must be unequivocally confirmed and corrected throughout the text and figure to maintain data integrity.

2. Electrophysiological Characterization of Cholinergic Interneurons (CINs). The reported intrinsic hyperexcitability of striatal CINs is a fascinating finding. However, the characterization provided is somewhat superficial. To robustly support this conclusion and allow the reader to fully assess the identity and health of the recorded cells, a more complete dataset is necessary. The manuscript should include:

  • A full table of passive properties (Input resistance, Membrane capacitance, Membrane time constant) for both genotypes, even if no significant difference was found.

  • A full characterization of action potential (AP) properties beyond threshold: AP amplitude (from threshold and from baseline), AP half-width, and afterhyperpolarization (AHP) amplitude. This is essential for confirming the cell type identity based on established electrophysiological fingerprints.

  • A description of the firing patterns (e.g., initial adaptation, steady-state firing) in response to current steps would provide deeper insight into the excitability changes.
    The authors mention filling neurons with biocytin. If any morphological reconstructions were attempted, even qualitative examples would be invaluable to correlate with the functional changes.

3. Methodology Organization and Reproducibility:
The Materials and Methods section requires reorganization and additional detail.

  • Organization: The description of techniques is currently interwoven and repetitive (e.g., perfusion and sectioning are described multiple times). The section should be restructured into logical subsections (e.g., Histology & Immunostaining, Microscopy & Image Analysis, Western Blot, Behavioral Assays, Electrophysiology) to improve readability.

  • Antibody Information: For all immunohistochemistry and Western blot experiments, the article number for each primary and secondary antibody must be provided. This is a critical requirement for reproducibility.

Minor Points:

  • Figure 2c: The graph for "Total rearing time and events" currently uses a single scale. This should be clarified, ideally by using a double Y-axis or separating the data into two panels.

  • Figure 7i and 7l: For consistency and transparency, the plots for Rheobase (7i) and AP frequency @ 200pA (7l) should display individual data points, as is done correctly in other panels like 7f and 7j.

Author Response

Revisions Round 1 – 10/3/25 

We thank the reviewers and editors for their thoughtful and constructive feedback on our manuscript, “A cadherin mutation in Celsr3 linked to Tourette Disorder affects dendritic patterning and excitability of striatal cholinergic interneurons”. 

We have carefully revised the manuscript in accordance with the reviewers’ comments, ensuring that the revised portions are clearly highlighted, and suggested literature has been evaluated. Where appropriate, we have incorporated the reviewers’ recommendations, and in cases where a comment could not be addressed directly, we provide a detailed explanation in our responses below. We believe these revisions have substantially strengthened the clarity and impact of the manuscript, and we are grateful for the opportunity to improve our work. 

(All text changes based on reviewer comments are highlighted in yellow in the updated manuscript) 

Comments: 

Reviewer #2: 

  1. The authors focus their core analysis on homozygous Celsr3R774H/R774H mice. However, it is noted that human TD-associated mutations in CELSR3 are heterozygous. While the rationale for using homozygotes (stronger phenotype, viability) is understandable, the relevance of the findings to the human heterozygous condition should be explicitly discussed as a potential limitation of the model. Furthermore, a critical inconsistency must be clarified: the Western blot in Figure 1b is described in the main text as comparing Celsr3+/+ and Celsr3R774H/R774H, yet the figure legend lists Celsr3R774H/+ (heterozygotes). The genotypes used in this crucial experiment must be unequivocally confirmed and corrected throughout the text and figure to maintain data integrity.  

Response: Thank you for correcting this error. The figure has been corrected (see figure 1b). The use of homozygous mutant animals in the western is clearly stated in the results section in lines 144-145. We also discuss this important caveat in lines 526-531. 

  1. Electrophysiological Characterization of Cholinergic Interneurons (CINs). The reported intrinsic hyperexcitability of striatal CINs is a fascinating finding. However, the characterization provided is somewhat superficial. To robustly support this conclusion and allow the reader to fully assess the identity and health of the recorded cells, a more complete dataset is necessary. The manuscript should include: 
  • A full table of passive properties (Input resistance, Membrane capacitance, Membrane time constant) for both genotypes, even if no significant difference was found. 
  • A full characterization of action potential (AP) properties beyond threshold: AP amplitude (from threshold and from baseline), AP half-width, and afterhyperpolarization (AHP) amplitude. This is essential for confirming the cell type identity based on established electrophysiological
  • A description of the firing patterns (e.g., initial adaptation, steady-state firing) in response to current steps would provide deeper insight into the excitability changes. 
    The authors mention filling neurons with biocytin. If any morphological reconstructions were attempted, even qualitative examples would be invaluable to correlate with the functional changes.  

Response:  

  • A table has been added with the requested info regarding electrophysiology experiments. Please see appendix A3.
  • We confirmed that the neurons used for recordings were in the dorsolateral striatum, were GFP+, and had the correct morphology through filling with biotin (please see updated figure panel 7B-D).  Representative tracings of striatal cholinergic interneurons have been added to figure 7a as qualitative examples. Additionally, we provide a quantitative assessment of changes to dendritic patterning in Supplemental Fig. 2. We also added data to suggest spine density is altered on secondary dendrites of CINs in mutant mice (Fig. 7 m, n). We initially chose to omit this data given that the number of cells analyzed is skewed towards mutants (n=6 controls, n=13 mutants). This is because several of the filled WT cells ultimately did not pass quality control due to breaks in the dendrtic branches (presumably because these processes were not fully captured in thick sections) or incomplete filling. Nonetheless, the findings, as presented, are interesting and may explain, at least in part, observed changes to the membrane properties of CINs.

  1. Methodology Organization and Reproducibility: 
    The Materials and Methods section requires reorganization and additional detail. 
  • Organization: The description of techniques is currently interwoven and repetitive (e.g., perfusion and sectioning are described multiple times). The section should be restructured into logical subsections (e.g., Histology & Immunostaining, Microscopy & Image Analysis, Western Blot, Behavioral Assays, Electrophysiology) to improve readability. 

Response: As also suggested by reviewer 1, we reorganized the methods section in light of the comments. 

  • Antibody Information: For all immunohistochemistry and Western blot experiments, the article number for each primary and secondary antibody must be provided. This is a critical requirement for reproducibility. 

Response: Catalog numbers have been added for all primary and secondary antibodies within       the materials and methods section.  

Minor Points: 

  • Figure 2c: The graph for "Total rearing time and events" currently uses a single scale. This should be clarified, ideally by using a double Y-axis or separating the data into two panels.  

Response: Thank you. We have changed figure 2c and added a separate y-axis for each rearing time and rearing count.  

  • Figure 7i and 7l: For consistency and transparency, the plots for Rheobase (7i) and AP frequency @ 200pA (7l) should display individual data points, as is done correctly in other panels like 7f and 7j.  

Response: We have changed figure 7f and 7j to display individual data points.  

Round 2

Reviewer 1 Report

Comments and Suggestions for Authors

As attached.

Author Response

4c-f. Given access to Imaris, it should be feasible to report the interneuron/reporter colocalization rate and interneuron density.

Response: We apologize if we misunderstand the critique, but we did not use IMARIS because nuclear markers provide sufficient resolution for quantifying co-localization between ChAT/PV staining and GFP. Of note, the average distancebetween cholinergic interneurons was quantified using IMARIS and nearest neighbor analysis, revealing no differences in cell density or distribution.

My point was that since you have access to Imaris, quantifying co-localization should be quite straightforward. If your nuclear markers provide sufficient resolution for co- localization analysis, you can simply import the mergedfiles into Imaris and use the Spot module to quickly obtain the co-localization ratio within specific laminar regions. This would greatly enhance the scientific soundness of your results. The sentence you mentioned “The averagedistance between cholinergic interneurons was quantified using IMARIS and nearest neighbor analysis, revealing no differences in cell density or

distribution.” is not directly related to whether these cells are differentially distributed across layers.

Response: Thank you for clarifying this point. While this is certainly a reasonable suggestion, we unfortunately do not have access to a paid IMARIS subscription at the present time due to funding constraints. Also, the cortical PVIN and SSTIN distribution quantifications we presented in figure 4 use different methodologies; for PVINs we used an immunostaining approach on tissue collected from Celsr3-GFP mice, whereas SSTINs were counted in “SST-cre; Ai14” double transgenic mice, so colocalization with Celsr3-GFP would not be possible for SSTINs. Of note, cholinergic interneurons are not found in the cortex, so it would not be possible to discern changes in distribution/density in different cortical layers.

  1. The cellular sample size is small, and the number of mice analyzed is not reported; thus, it is impossible to evaluate how much the result is driven by between-subject differences.

We analyzed neurons from 3 different mice of each genotype. Due to the nature of the sparse-cell labeling, the total number of neurons analyzed is somewhat limited. That being said, the differences were striking between mutant and control neurons, despite the limited data set, and agree with previous observations noted in Celsr3 conditional KO mutants that we cite in the discussion.

I understand that it’s difficult to collect enough cells because they are sparsely distributed. However, pleasemake sure to clearly state in the figure legend or the main text that you analyzed neurons from three different mice for each genotype.

Response: Thank you, we have now clarified the information regarding the sample sizes in the figure legend.

  1. Two animals per group is insufficient for inferential statistics; please increase the number of animals before drawing conclusions.

Response: We analyzed another animal and now have an n=3 for each group.

Since you already have n = 3, perhaps you could perform a Wilcoxon signed-rank test to statistically verify whether the difference is significant.

Response: We performed a Wilcoxon signed-rank test on this data. The difference failed to reach significance, but given the small sample size, the test is underpowered to detect small effect sizes. However, it was not our specific intention to derive statistical differences in GFP reporter expression among different striatal cell types. Rather, we presented this data as logic for choosing to interrogate Celsr3 functions in cholinergic interneurons (vs. PV interneurons), based upon more widespread GFP expression in these cell types.

Reviewer 2 Report

Comments and Suggestions for Authors

The authors answered all my questions, I have no further comments.

Author Response

We thank Reviewer 2 for their thoughtful evaluation and constructive suggestions, which have helped us to significantly improve the clarity and quality of our manuscript.